# Asymmetric gene expression in grain development of reciprocal crosses between tetraploid and hexaploid wheats

Zhen Jia[1,6], Peng Gao [2,6], Feifan Yin[1,3,6], Teagen D. Quilichini[4], Huajin Sheng[2], Jingpu Song[4], Hui Yang[4], Jie Gao[1], Ting Chen[1], Bo Yang[1], Leon V. Kochian[2], Jitao Zou [4], Nii Patterson[4], Qingyong Yang[1,3], C. Stewart Gillmor[5], Raju Datla[2✉], Qiang Li[1✉] & Daoquan Xiang [4✉]

Production of viable progeny from interploid crosses requires precise regulation of gene expression from maternal and paternal chromosomes, yet the transcripts contributed to hybrid seeds from polyploid parent species have rarely been explored. To investigate the genome-wide maternal and paternal contributions to polyploid grain development, we analyzed the transcriptomes of developing embryos, from zygote to maturity, alongside endosperm in two stages of development, using reciprocal crosses between tetraploid and hexaploid wheats. Reciprocal crosses between species with varied levels of ploidy displayed broad impacts on gene expression, including shifts in alternative splicing events in select crosses, as illustrated by active splicing events, enhanced protein synthesis and chromatin remodeling. Homoeologous gene expression was repressed on the univalent D genome in pentaploids, but this suppression was attenuated in crosses with a higher ploidy maternal parent. Imprinted genes were identified in endosperm and early embryo tissues, supporting predominant maternal effects on early embryogenesis. By systematically investigating the complex transcriptional networks in reciprocal-cross hybrids, this study presents a framework for understanding the genomic incompatibility and transcriptome shock that results from interspecific hybridization and uncovers the transcriptional impacts on hybrid seeds created from agriculturally-relevant polyploid species.

[1] National Key Laboratory of Crop Genetic Improvement, Huazhong Agricultural University, 430070 Wuhan, China. [2] Global Institute for Food Security, University of Saskatchewan, Saskatoon, SK S7N 4J8, Canada. [3] Hubei Key Laboratory of Agricultural Bioinformatics, College of Informatics, Huazhong Agricultural University, 430070 Wuhan, China. [4] Aquatic and Crop Resource Development, National Research Council Canada, 110 Gymnasium Place, Saskatoon, SK S7N 0W9, Canada. [5] Langebio, Unidad de Genómica Avanzada, Centro de Investigación y Estudios Avanzados del IPN (CINVESTAV-IPN), Irapuato, Guanajuato 36821, México. [6] These authors contributed equally: Zhen Jia, Peng Gao, Feifan Yin. ✉email: raju.datla@gifs.ca; qli@mail.hzau.edu.cn; daoquan.xiang@nrc-cnrc.gc.ca

Hybridization of tetraploid durum wheat (*Triticum turgidum ssp.* durum; $2n = 4x = 28$) and hexaploid bread wheat (*Triticum aestivum* L.; $2n = 6x = 42$) produces pentaploid hybrids with improved agronomic characters including disease resistance and abiotic stress tolerance[1–4]. These improvements result from extensive genetic variation associated with the predominance of heterozygous loci in the A and B genomes, together with the retention of a haploid D genome. However, polyploid hybridization between different wheat species can yield incompatibility and sterility issues that challenge trait introgression efforts. For example, pentaploid wheats with retained D genome chromosomes or both A and B genomes have resulted in growth abnormalities and varied fertility in the F2 and subsequent generations[5,6]. Pentaploid interspecific hybrids between wild emmer and common wheat cultivars produced highly sterile offspring (fertility of l%–2%)[7], although such fertility issues can usually be overcome with further crosses between the hybrids and hexaploid wheat cultivars to hasten the recovery of euploid progeny ($2n = 42$) with introgressed genes[5,8].

Studies on interspecific hybridization of hexaploid and tetraploid wheat have found that the use of higher ploidy species as the maternal parent improves seed set and germination[5,6,8]. Most interspecific hybridizations between different ploidy levels use the hexaploid wheat as the female parent[5,8,9]. Although an interspecific wheat cross using the tetraploid durum wheat as the female parent and the hexaploid bread wheat as the male parent has been developed, none of the F2 progeny contained a complete set of the seven D genome chromosomes[9]. Therefore, understanding the barriers to hybridization and cross-directional effects is critical for developing successful interspecific wheat hybrids.

In plants, interspecific hybridization can lead to genomic accommodations such as chromosome reconstitution[10], nuclear-cytoplasmic interactions[11], non-additive gene expression, altered alternative splicing (AS)[12], and changes in epigenetic regulation[13]. Changes in gene expression and genome dominance have been observed in hybridization and polyploidization events in plants and animals[14–16]. The reunion of two diverged genomes into a common nucleus during hybridization can result in negative or additive effects on transcript levels in different tissues, growth stages and response to abiotic stresses[17,18]. For example, transcriptome analyses in seedlings of maize revealed alterations in the expression of a number of genes in F1 hybrids compared to their inbred parents[19]. Moreover, additive gene expression patterns and biased allele abundance appear to be prevalent in maize hybrids[19]. In allopolyploid species such as cotton and wheat, hybridization often leads to imbalances in homeolog gene expression and homeolog silencing, derived from inter- and intra-subgenome interactions[15,20,21]. Homeolog expression bias is proposed to be regulated by *cis*- and *trans*-acting regulatory interactions among genes and alleles in hybrids and allopolyploids[22]. However, the effects of merging two regulatory networks into one genetic system and the impact on genome-wide allelic expression changes in polyploidy species remain largely unexplored.

Here, we present a comprehensive gene expression study of reciprocal-cross hybrids of tetraploid and hexaploid wheat species during grain development, with a focus on the reprogramming of gene expression and AS in the embryo. We hypothesize that allelic imbalance and asymmetric expression of maternally and parentally derived genomes represent alternative strategies to counteract deleterious effects of merging subgenomes and to improve the adaptability of new hybrids. This study provides insights into the interspecific hybridization associated with the evolution of gene expression in polyploid species during embryogenesis and grain development in wheat.

## Results

**Transcriptomes of developing grains in reciprocal crosses of polyploid wheats.** To monitor the reprogramming of transcriptional activities during embryo development in polyploid reciprocal crosses, two hexaploid wheat varieties, Chinese Spring (hereafter called Z) and AC Barrie (hereafter called A) and two tetraploid varieties, Strong Field (hereafter called S) and Commander (hereafter called C) were selected (Fig. 1a). Reciprocal crosses were performed between varieties with different ploidy levels (between hexaploids and tetraploids) and the same ploidy levels (crossing hexaploids or crossing tetraploids) (Fig. 1a). Following each of the four reciprocal crosses, the embryo and endosperm were isolated from the F1 progeny and RNA was extracted from each grain component. The developing embryos and endosperm resulting from each reciprocal cross did not exhibit notable morphology differences at collected stages (Supplementary Fig. 1a). However, statistical analysis of mature F1 seeds of different crosses showed the seed weight, seed length, seed width, seed perimeter and seed area were significantly reduced in tetraploid × hexaploid and hexaploid × tetraploid interspecific crosses, compared to the same ploidy crosses (Supplementary Fig. 1b, c). In particular, fewer and some abnormal endosperm cells were observed in tetraploid × hexaploid and hexaploid × tetraploid interspecific crosses, when compared to the endosperm resulting from the same ploidy crosses (Supplementary Fig. 1b), indicating endosperm cell division, endosperm development and seed starch accumulation are impacted after interspecific hybridization. Furthermore, the grain weight always showed higher values both in the interspecific crosses and the same ploidy crosses if the maternal parent has larger seed size than paternal parent in mature seeds (Supplementary Fig. 1b), suggesting an maternal impact on grain development after hybridization. RNA sequencing of samples from sequential stages of grain development included seven stages of embryo development (from zygote to mature embryo, E1 to E7), and two stages of endosperm development (early and late stage, E8 and E9) (Fig. 1a, b). We deep sequenced 144 RNA samples for developing embryos and endosperms derived from reciprocal crosses of polyploid wheats, as summarized in Supplementary Data 1.

Investigating transcriptional dynamics across multiple samples with different ploidy levels requires the construction of a consensus transcriptome. To obtain a high-confidence reference-based annotation based on multi-sample RNA-seq data, we tested different assembly methods to annotate high-fidelity gene models. For hexaploid (AABBDD) and pentaploid (AABBD) samples, the raw RNA-seq sequence reads were aligned to the entire IWGSC RefSeq 1.1 genome assembly. The tetraploid samples (AABB) were aligned to the A and B subgenomes. The combined initial mapping results were summarized, and an average of 93.3% of reads were successfully aligned (Supplementary Data 2). The genome-guided transcript assemblies served as inputs for software-guided transcript identification (Supplementary Fig. 2). Performance of nine methods was assessed for the sensitivity and precision of transcript identification. Method 2 had the highest sensitivity and precision, and was used for subsequent analyses (see details in Supplementary Fig. 2).

Among the 133,363 identified genes, 87,568 had transcript per million (TPM) counts greater than one across all samples (Supplementary Data 3). The total number of expressed genes detected at each stage (TPM > = 1) was relatively steady as development proceeded (Supplementary Fig. 3). A correlation matrix comparing sample expression levels for all genes revealed distinct expression profiles for samples of different ploidy levels, with hexaploids exhibiting a higher correlation with pentaploids than with tetraploids. In addition, closely correlated gene expression profiles were observed between neighboring stages (Fig. 1c).

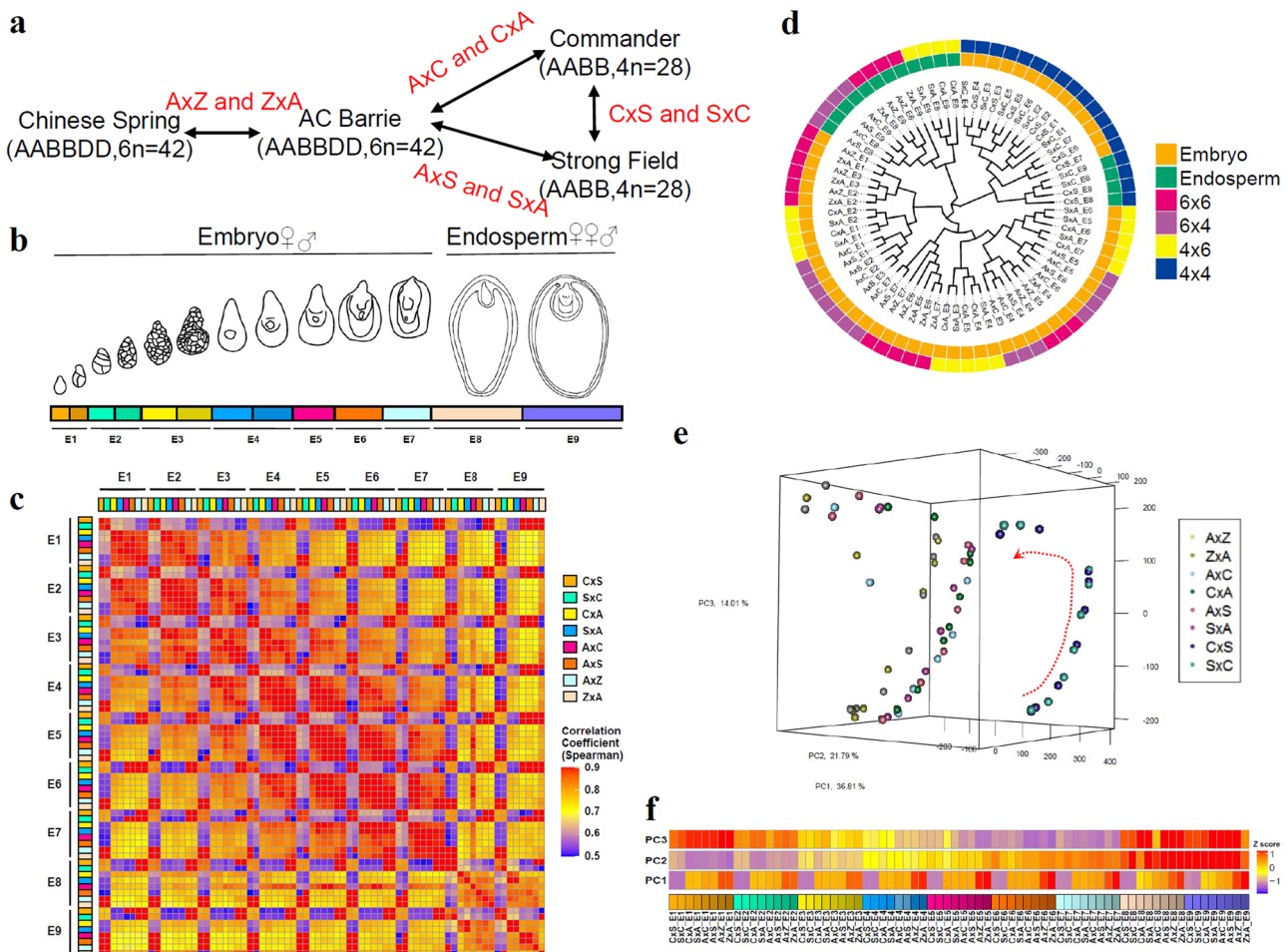

**Fig. 1 Transcriptome-wide landscape of gene expression profiles during embryogenesis after hybridization between polyploid wheat species.**
**a** Experimental design and interspecific hybridizations. Reciprocal crosses were conducted between hexaploids, between hexaploid and tetraploid and between tetraploids. A = AC Barrie; C = Commander; S = Strong Field; Z = Chinese Spring. **b** Sampling stages of embryo (E1 to E7) and endosperm (E8 and E9) in this study. **c** Sample correlation matrix of reciprocal samples. Spearman correlation coefficients were calculated using normalized counts of all identified genes. **d** Phylogenetic tree of all samples. Different tissues (embryo and endosperm) and crosses between different ploidy levels are shown. 6 × 6, cross between hexaploids; 6 × 4, cross between hexaploid and tetraploid using hexaploid as female parent; 4 × 6, cross between tetraploid and hexaploid using tetraploid female parent; 4 × 4, cross between tetraploids. **e** 3D plot of principal component analysis (PCA) showing the first three principal components (PCs), which together explain 72.6% of the variance. Samples from the same stage cluster together and there is a smooth progression through developmental time (shown as in a red dashed arrow). Samples with different ploidy levels were separated. The amount of variance explained by each PC is indicated on each axis. **f** Representation of the expression profiles that contribute most to the first 3 PCs. The expression values are centered and scaled (Z-score) for all genes that contribute most to that PC and then the mean value for each sample is plotted. C×S, Commander(♀) × Strong Field(♂); S×C, Strong Field (♀) × Commander(♂); C×A, Commander(♀) × AC Barrie(♂); A×C, AC Barrie(♀) × Commander(♂); S×A, Strong Field(♀) × AC Barrie(♂); A×S, AC Barrie(♀) × Strong Field(♂); Z×A, Chinese Spring(♀) × AC Barrie(♂); A×Z, AC Barrie(♀) × Chinese Spring(♂).

A phylogenetic tree separated reciprocal samples into two groups (Fig. 1d). Similar to the correlation matrix, hexaploid and pentaploid samples were grouped together, apart from tetraploid samples. Embryo and endosperm samples were also separated. Endosperms of hexaploid and pentaploid samples grouped together and apart from the endosperm tetraploid sample cluster, suggesting that ploidy level differences have a larger effect than tissue differences.

Principal component analysis (PCA) showed a smooth transition in sequential developmental stages throughout embryogenesis, with endosperm samples clearly separated from embryos (Fig. 1e and Supplementary Fig. 4). A summary of the expression patterns underlying the principal components (PCs) is presented in Supplementary Fig. 4. The first three PCs account for 72.61% of the observed variance. PC1 explains ~36.81% of the variance, separating the reciprocal crosses into hexaploid, tetraploid and pentaploid samples (Fig. 1e and Supplementary Fig. 4). PC2 (21.79% of variance) and PC3 (14.01%) explain the variance

amongst developmental stages and tissues, respectively. In general, genes that contribute the most to the first PC are expressed at lower levels in tetraploids and increase along with ploidy levels. Genes contributing to PC2 are lower at early embryonic stages and increase towards the mature stage embryos or endosperm. Genes contributing to PC3 have two expression peaks, one in early stage embryos and another in endosperm tissue (Fig. 1f).

**Differential gene expression patterns in progressive developmental stages of wheat embryogenesis**. We first evaluated differential expression in each cross during embryo development. Genes with more than one TPM in at least one of all samples were considered expressed. Differentially expressed genes (DEGs) in each embryo sample were identified through comparisons with the two-cell stage (E1). Genes with an adjusted $p$-value < 0.01 and log2fold-change >= 1 or =< −1 in at least one sampling time

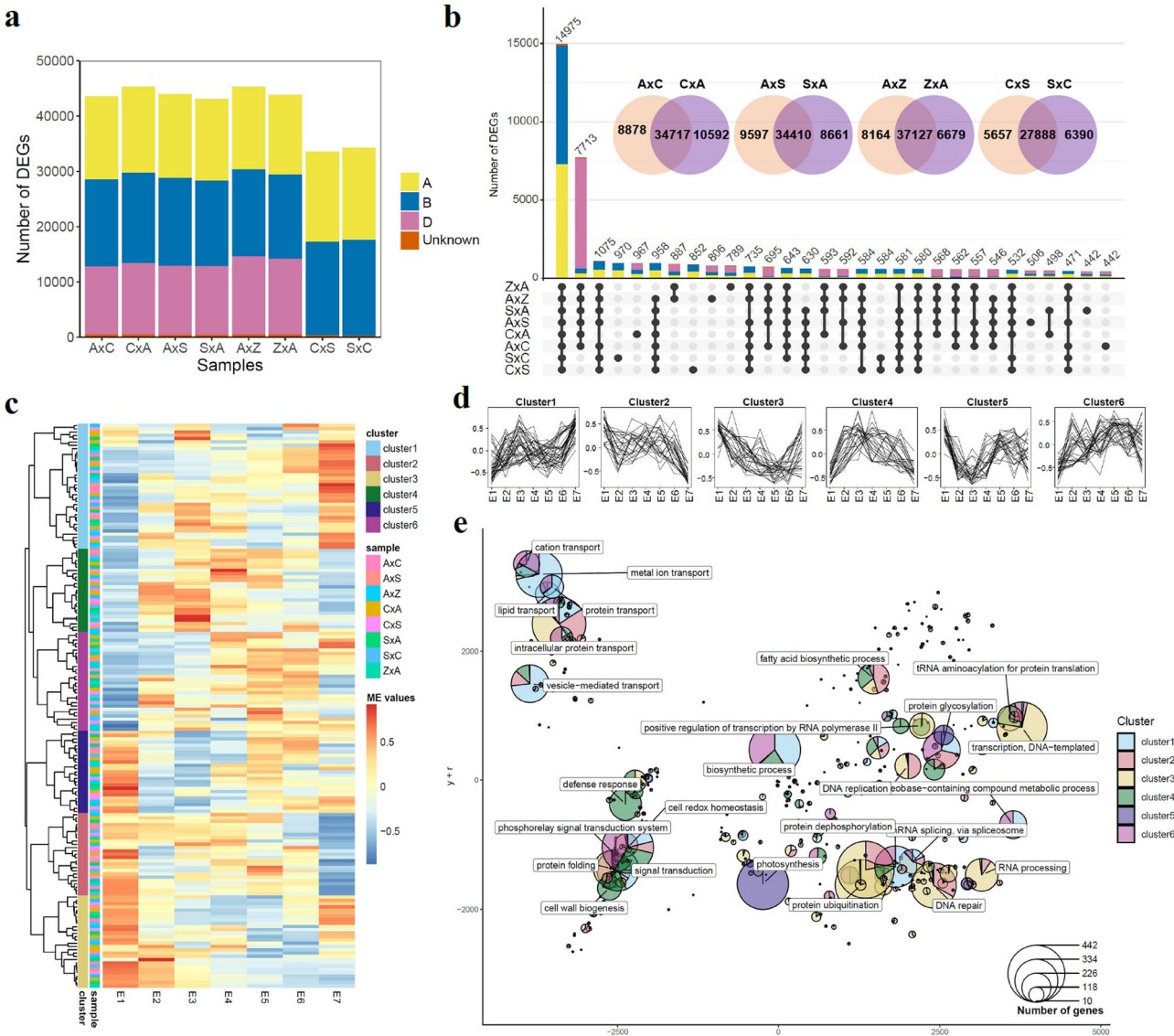

**Fig. 2 Differential expression analysis and clustering of expression patterns across multiple samples. a** The number of differentially expressed genes (DEGs) in each sample. DEGs were identified through comparisons between each time point against the two-cell stage for each sample. **b** Number of DEGs shared between crosses as shown by UpSet plots and Venn diagrams. **c** Clustering of module eigengene values (MEs). MEs represent the first principal component of each WGCNA module(or cluster) in each of the eight reciprocal cross samples. A total of 170 clusters were identified and ME values of each cluster were hierarchically clustered. **d** Expression pattern of six distinct clusters identified from hierarchical clustering. **e** Scatterpie chart showing the significantly enriched GO terms of each cluster. GO terms were slimmed by grouping similar terms based on their semantic similarity. Each circle in the pieplot represents a GO term in biological processes. Distances between each circle were determined by their GO similarity and axes are the first two components of applying a PCA to the similarity matrix. Size of each circle represents the total number of genes in the GO term. Proportion of genes from different clusters in each circle were colored as shown.

point were considered differentially expressed (Fig. 2a and Supplementary Data 4). A total of 64,965 genes were identified as embryo DEGs (Supplementary Data 4), and the number of DEGs identified increased as development progressed (Supplementary Fig. 5). Over 40,000 DEGs were identified in hexaploid and pentaploid samples, and tetraploid samples contained ~30,000 DEGs (Fig. 2a). To visualize the unique and common DEGs, intersections between reciprocal crosses were identified (Fig. 2b). More than 60% of DEGs were shared between each reciprocal cross pair. A total of 14,975 DEGs were commonly identified in all samples and 7713 were shared by hexaploids and pentaploids (Fig. 2b). To identify gene expression patterns during embryogenesis, we performed cluster analysis for each sample (Supplementary Figs. 6–13). Weighted correlation network analysis

(WGCNA) identified co-expressed genes from the expression profiles of all samples simultaneously, using a dynamic hierarchical clustering approach. A total of 170 clusters were identified in the eight reciprocal crosses and functional enrichment for each of the clusters was performed (Supplementary Fig. 14).

Direct comparison of gene expression patterns and enriched biological functions across species with different ploidy levels remains a major challenge, especially for studies examining numerous samples[23]. To compare gene expression patterns between samples, we calculated module eigengenes (MEs), which represent the first principal component of each WGCNA module in each sample (Supplementary Figs. 6–13). ME values were hierarchically clustered and six distinct ME expression patterns were identified across all samples (Fig. 2c, d). To facilitate biological interpretation,

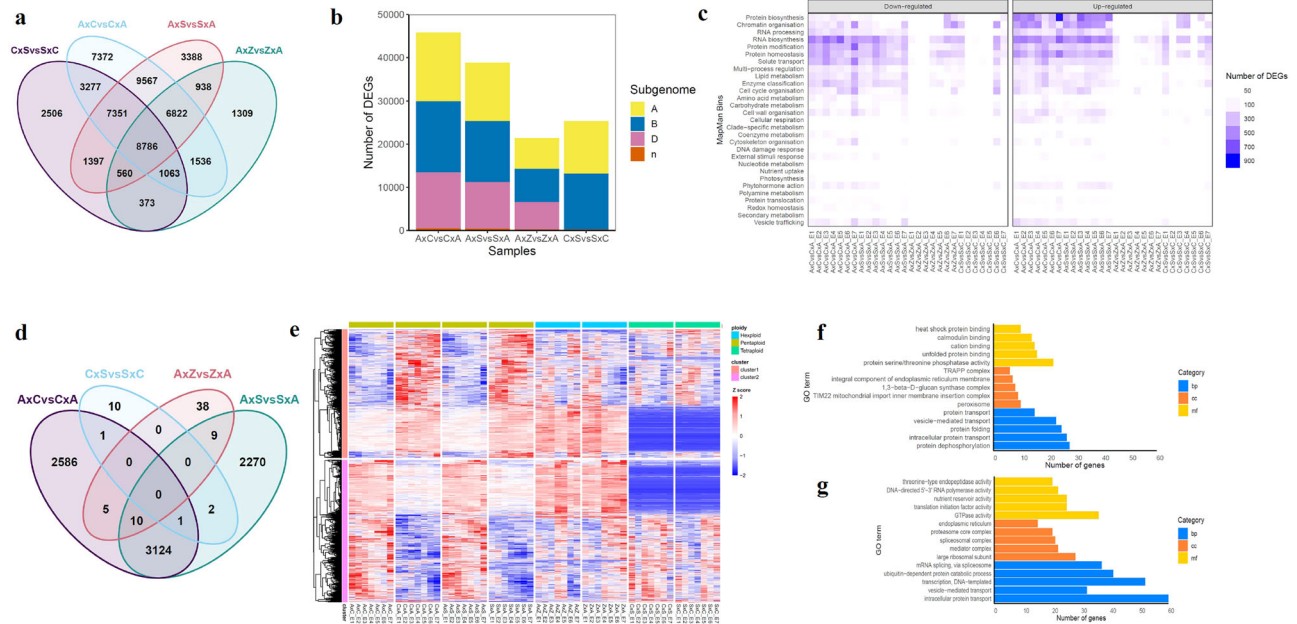

**Fig. 3 Differential expression analysis between reciprocal cross pairs. a** Venn diagram of DEGs between reciprocal cross pairs. **b** Number of DEGs between each of the reciprocal cross pair. **c** Number of DEGs in different pathways. MapMan bins of genes were annotated using the Mercator pipeline (http://www.plabipd.de/portal/mercator-sequence-annotation)[70]. The number of up- and downregulated DEGs between reciprocal pairs in each pathway were plotted. Upregulated DEGs represent genes that are expressed at higher levels in A×C against C×A, A×S against S×A, A×Z against Z×A and C×S against S×C. **d** Differentially expressed genes across all embryo stages between each reciprocal pair. ANOVA tests were performed for each DEGs to identify genes that are consistently highly or lowly expressed across seven embryo stages between each reciprocal pair (FDR < 0.05). Heatmap (**e**) and GO enrichment analysis of genes showing different expression patterns in cluster 1 (**f**) and cluster 2 (**g**) between reciprocal cross pairs across seven embryo stages. Normalized expression is scaled across genes (Z-score) and colored in heatmap.

multiple enriched GO terms were organized according to the dendrogram of ME clustering (Supplementary Fig. 14). Clusters with similar trends for expression patterns tended to have similar biological functions (Fig. 2e and Supplementary Fig. 14), allowing conserved gene expression patterns across embryo development to be identified in reciprocal crosses. For example, upregulated genes in cluster 1 were predominantly associated with cell redox homeostasis, metal ion transport, biosynthetic process and vesicle-transport. Downregulated genes along the embryo development stages in cluster 2 were involved in DNA repair, DNA replication, RNA processing, protein ubiquitination and photosynthesis. Embryo gene expression showed similar changes in all samples as development progressed (Supplementary Fig. 15 and Supplementary Data 5). For example, relative to E1, the expression of three genes (*TraesCS2A02G441200*, *TraesCS2B02G147500* and *TraesCS2D02G437700*) encoding ubiquitin extension protein 1 (EMB2167 or UBQ1), a protein known to be involved in embryo development[24], had decreased expression in all samples, relative to E1. In contrast, the expression of *TraesCS4A02G114200*, *TraesCS4B02G189900* and *TraesCS4D02G191300*, which encode malonyl CoA-ACP malonyltransferase (MCAMT or EMB3147), responsible for generating the malonyl-acyl carrier protein for fatty acid biosynthesis[25], was upregulated across all stages of embryogenesis relative to E1.

**Transcriptional changes between reciprocal crosses during embryogenesis.** To assess the impacts of cross-directional crosses (hexaploid x tetraploid vs. tetraploid x hexaploid) on global gene expression, we identified DEGs in four pairs of reciprocal crosses. A total of 45,774, 38,809, 21,387, and 25,313 DEGs were identified in embryos from A×C vs. C×A; A×S vs. S×A; C×S vs. S×C; and A×Z vs. Z×A crosses, respectively (Fig. 3a and Supplementary Data 6). Across seven embryo stages, more DEGs were identified in

reciprocal crosses between tetraploid and hexaploid species (A×C vs. C×A and A×S vs. S×A) than in crosses between wheats with the same ploidy levels (Fig. 3b). A total of 9567 DEGs were uniquely identified in two pairs of pentaploids, respectively (Supplementary Fig. 16). The DEGs identified between reciprocal crosses are indicative of parental effects on gene expression during embryogenesis.

DEGs were further categorized into upregulated and downregulated genes. Upregulated DEGs represent genes that are expressed at higher levels in A×C/C×A, A×S/S×A, A×Z/Z×A, and C×S/S×C. Large differences in enriched GO terms were found between reciprocal crosses with different ploidy levels (A×C vs. C×A and A×S vs. S×A) when compared to crosses within the same ploidy levels (A×Z vs. Z×A and C×S vs. S×C). DEGs in A×C vs. C×A and A×S vs. S×A comparisons were enriched in similar GO terms (Supplementary Fig. 17). For example, downregulated genes were enriched in biological processes such as microtubule-based movement, signal transduction, protein ubiquitination, protein folding, and protein transport, while upregulated genes were enriched in cell redox homeostasis, fatty acid biosynthetic process, ribosome biogenesis, intracellular protein transport, mRNA splicing via spliceosome, DNA-templated transcription, vacuolar transport, translational elongation and protein transmembrane transport (Supplementary Fig. 17). Distinct pathways were affected after hybridization between different ploidy levels (Fig. 3c). DEGs in pentaploids were mainly involved in protein biosynthesis, protein modification, protein homeostasis, RNA biosynthesis and chromatin organization. It should be noted that a number of genes involved in protein biosynthesis and chromatin organization were upregulated in A×C and A×S compared to C×A and S×A.

We performed paired ANOVA tests for each DEG to identify genes consistently expressed at higher or lower levels across all embryo stages between each reciprocal pair. A total of 5727 and 5416 genes were identified in A×C vs. C×A and A×S vs. S×A,

respectively (FDR < 0.05). Of these genes, 3124 genes were commonly identified in the pentaploids, while in A×Z vs. Z×A and C×S vs. S×C where parents have the same ploidy levels, only 62 and 14 genes were identified (FDR < 0.05) (Fig. 3d). These genes were further categorized into two distinct expression patterns (Fig. 3e and Supplementary Fig. 18). Cluster 1 had genes expressed at higher levels in pentaploids whose female parent is tetraploid, while cluster 2 contained genes with the opposite expression pattern, with expressed at lower levels in pentaploids with a tetraploid female parent. GO enrichment showed that genes in cluster 1 were enriched in protein transport, protein folding, and protein dephosphorylation (Fig. 3f). Genes in cluster 2 were involved in mRNA splicing via spliceosome, intracellular protein transport, vesicle-mediated transport, and transcription DNA-templated (Fig. 3g). Over 200 DEGs involved in embryo development[26] were identified in pentaploids, while less than 100 DEGs were identified in hexaploids and tetraploids (Supplementary Fig. 19 and Supplementary Data 7). Some genes were consistently highly expressed in pentaploids whose female parent is hexaploid, including for example, *ubiquitin extension protein 1* (*EMB2167* or *UBQ1*) and *ribosomal protein 5A* (*RPS5A*)[24,27].

Together, our results demonstrate extensive changes in transcriptional regulations in F1 hybrid embryos after hybridization, and to a greater extent in crosses between wheats with different ploidy levels. Moreover, analysis of cross-directional effects suggests alternative splicing, protein processing and chromatin remodeling are important processes in embryogenesis in pentaploids with a hexaploid female parent, while protein modification and transport are involved in regulating embryogenesis in pentaploids with a tetraploid female parent.

**AS divergence in reciprocal-cross hybrids**. Transcripts from the same gene can be altered through AS to create mature mRNAs with different nucleotide sequences and protein products. The transcript-specific data generated in this study allowed changes in AS during embryogenesis to be captured. Percent inclusion, or percent spliced-in (PSI, the ratio between reads including or excluding exons) has been widely used to evaluate splice junction sites[28]. Changes in PSI values indicate a shift in splicing pattern between AS events. We quantified PSI, the ratio of a transcript element over the total normalized reads for an AS event, using multiple stages of embryo development. AS analysis was performed with SUPPA v2.3[29]. To determine PSI for a gene, RNA-seq reads were mapped to the reference IWGSC 1.1 junction database and transcripts with $0.05 \le PSI \le 0.95$ were considered alternative. Consistent with previous studies[12,30], more than 50% of genes had one transcript (Supplementary Fig. 20 and Supplementary Data 8). We calculated possible local AS events including alternative 3′(A3) and 5′(A5) splice-site, intron retention (RI), exon skipping (SE), mutually exclusive exons (MX), alternative first exon (AF) and alternative last exon (AL). The number of AS events increased during development (Fig. 4a). More AS events were identified in late developmental stages (E4, E5, E6 and E7) than in early stages (E1, E2, and E3). Across all samples, the B genome showed the highest number of AS events, followed by the A and D genomes (Fig. 4b). The B genome had the highest ratio of RI events, while A3 type splicing events were dominant in the D genome (Fig. 4c). Although tetraploids showed a lower number of AS events than pentaploids and hexaploids, similar ratios of AS types were observed in reciprocal crosses at each stage (Fig. 4d).

Further analysis of PSI value distributions in reciprocal-cross hybrids revealed a similar distribution of PSI among pairs of reciprocal crosses, with most values close to 0 or 1, suggesting one transcript per gene (Fig. 4e). The distribution of PSI was significantly different between reciprocal crosses with parents

from different ploidy levels (Kolmogorov–Smirnov test, *p*-value < 0.01). Specifically, more AS events showed a PSI from 0.25 to 0.75 in samples with hexaploid species as female parent (A×C and A×S) than in samples with a tetraploid female parent (C×A and S×A) (Fig. 4e). In other words, more transcript elements or splicing events were found in A×C and A×S compared to C×A and S×A. Our results thus indicate a shift in splicing patterns in reciprocal-cross hybrids with different parental ploidy levels.

To explore the AS patterns caused by interspecific hybridization and differential AS (DAS), the magnitude of splicing change (ΔPSI) and its significance across multiple embryo stages was determined using SUPPA analysis[31]. DAS values were identified through pairwise comparisons between reciprocal crosses (|ΔPSI| > 0.1 and FDR < 0.05). During embryogenesis, crosses with parents of different ploidy levels produced more DAS than crosses with parents of the same ploidy levels (Fig. 4f and Supplementary Data 9). Since DAS represents differences in abundance of a transcript isoform between samples, we summed the number of upregulated or downregulated AS events between reciprocal crosses in all stages. Strikingly, the number of upregulated transcript isoforms was almost two times that of downregulated transcript isoforms in A×C vs. C×A and A×S vs. S×A reciprocal crosses. In contrast, a similar number of upregulated and downregulated transcript isoforms were identified in A×Z vs. Z×A and C×S vs. S×C (Fig. 4f).

To investigate whether genes involved in AS regulation were altered, we identified a total of 2039 genes involved in splicing regulation in wheat (Supplementary Data 10)[32,33]. Higher numbers ($n = 195-451$) of ribonucleoprotein (RNP) encoding genes were differentially expressed in A×C vs. C×A and A×S vs. S×A than in A×Z vs. Z×A and C×S vs. S×C ($n = 7-161$) (Fig. 4g). In addition, more upregulated RNPs were found in A×C vs. C×A and A×S vs. S×A. Splice regulator encoding genes such as *TraesCS7A02G304400* (*SmE*)[34], *XLOC_016378* (*LSM5*)[33] and *XLOC_165365* (*SmD*)[35] were upregulated in A×S and A×C relative to S×A and C×A, but remain unchanged in hexaploids and tetraploids. These data suggested that reciprocal crosses between species with different ploidy levels resulted in active AS and a notable shift in the expression pattern of transcript isoforms in F1 hybrids whose female parent has a higher ploidy level.

**Homoeologous gene expression in reciprocal-cross hybrids during embryogenesis**. Hexaploid bread wheat evolved through two interspecific hybridization steps involving three diploid donor species, each with seven pairs of chromosomes. As a result, allohexaploid wheat displays a high percentage of homoeologous genes in the A, B, and D subgenomes. In this study, 18,138 homoeologous pairs (also known as triads) corresponding to 54,414 genes with a 1:1:1 ratio across A, B, and D subgenomes were identified (Supplementary Data 11 and Supplementary Fig. 21). Analysis of synteny between triads in the three subgenomes showed high levels of conservation. Consistent with previous studies[36], rearrangements were found on chromosome 4A, 5A, and 7B (Supplementary Fig. 22).

A hallmark feature of polyploid genomes is the unequal contribution of A, B, and D subgenome homoeologs to total gene expression. While differential contributions from homoeologs have been observed between different tissues, stress conditions and developmental stages[15,37], little is known about the relative contributions of individual homoeologs to embryogenesis produced by reciprocal crosses. To compare the differential expression of homoeologs among the A, B, and D subgenomes in embryos, the relative contribution of each gene in each triad was determined. Genes in reciprocal crosses between the same

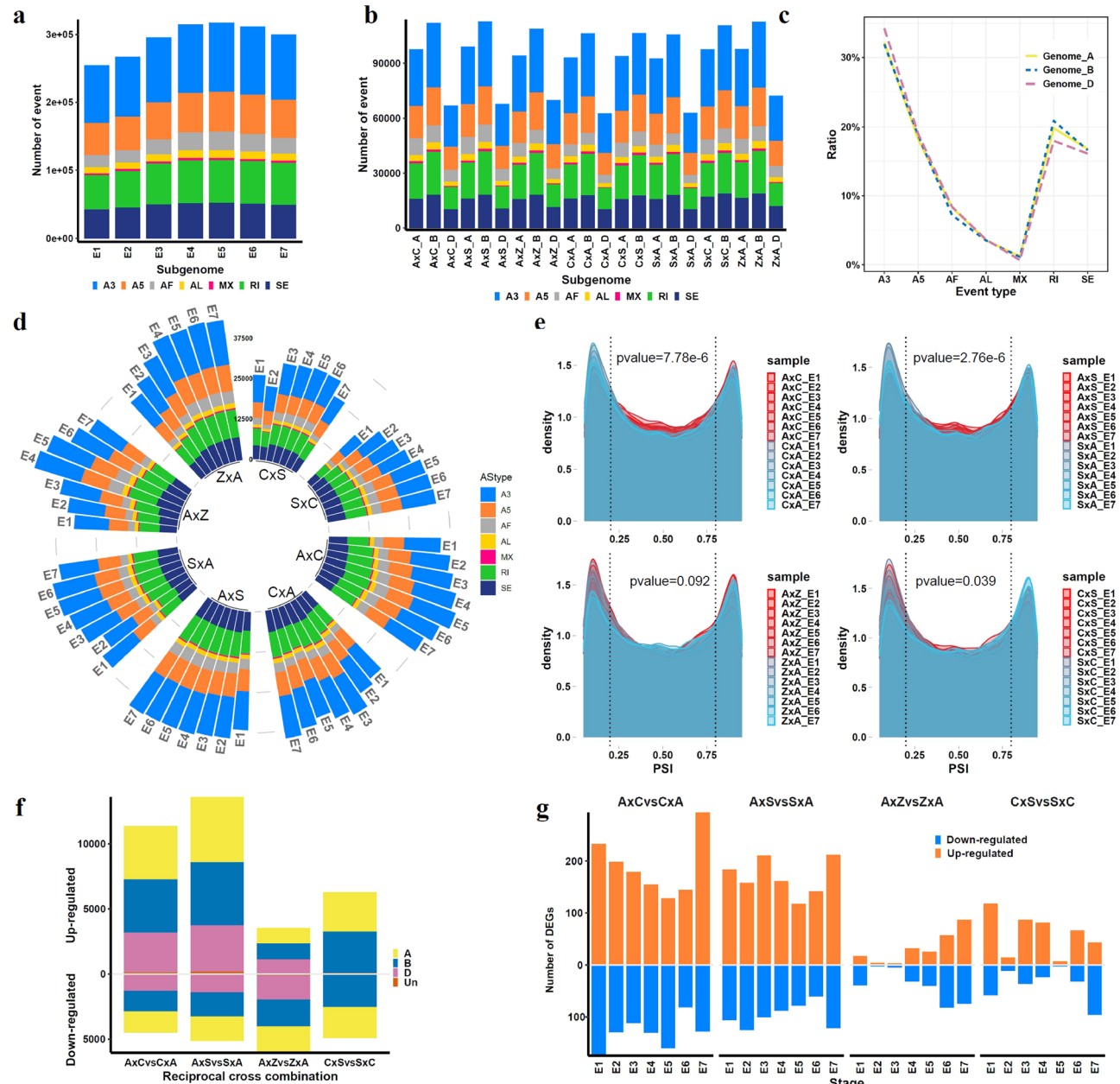

**Fig. 4 Interspecific hybridization resulted in altered AS pattern. a** Number of AS events identified at each stage in embryos. **b** Subgenome distribution of AS events in all embryo stages for each cross. **c** Ratio of each AS type on A, B, and D genome. **d** Number of AS events in reciprocal pairs. **e** Density of *psi* values in reciprocal pairs at embryo stages. Statistical significance between each cross pair is calculated by Kolmogorov–Smirnov test. **f** Number of differential AS between reciprocal cross pairs. A, A genome; B, B genome; D, D genome; Un, unknown. **g** Number of DEGs involved in splicing regulation between reciprocal pairs. Number of upregulated genes and downregulated genes involved in splicing pathways are shown.

ploidy levels showed the same gene expression patterns among the A, B, and D genomes (Fig. 5a). Balanced expression (defined as a 33% contribution from each gene in a triad) was observed in reciprocal crosses, including A×Z and Z×A. In C×S and S×C tetraploid crosses, the majority of homoeologous genes exhibited equal contributions from the A and B subgenomes. However, crosses between different ploidy levels resulted in an unequal contribution of homoeologs from the A, B, and D subgenomes, possibly due to the univalent D subgenome in pentaploids. Most genes from the D subgenome contributed 20% expression, while genes from the A and B subgenomes each contributed ~40% to total triad expression levels. These results support altered gene expression contributions in homoeologous triads produced by hybridization by parents of different ploidy levels.

DEGs, which are triads, were categorized into seven groups: balanced, A dominant, B dominant, D dominant, A suppressed, B suppressed and D suppressed (Fig. 5b). In hexaploid A×Z and Z×A crosses, more than 60% of homoeologs showed balanced expression, and similar proportions of A, B, or D genome dominant or suppressed expression were observed. In contrast, the pentaploids displayed unbalanced homoeologous expression patterns. In pentaploid embryos, there were 25–30% D genome suppressed homoeologs compared to less than 10% D genome suppressed homoeologs in hexaploid samples. In addition, more D genome suppressed homoeologs were found in S×A and C×A crosses with a tetraploid maternal parent than A×S and A×C crosses with a hexaploid maternal parent (Fig. 5b). These results support repression of homoeologous gene expression on the

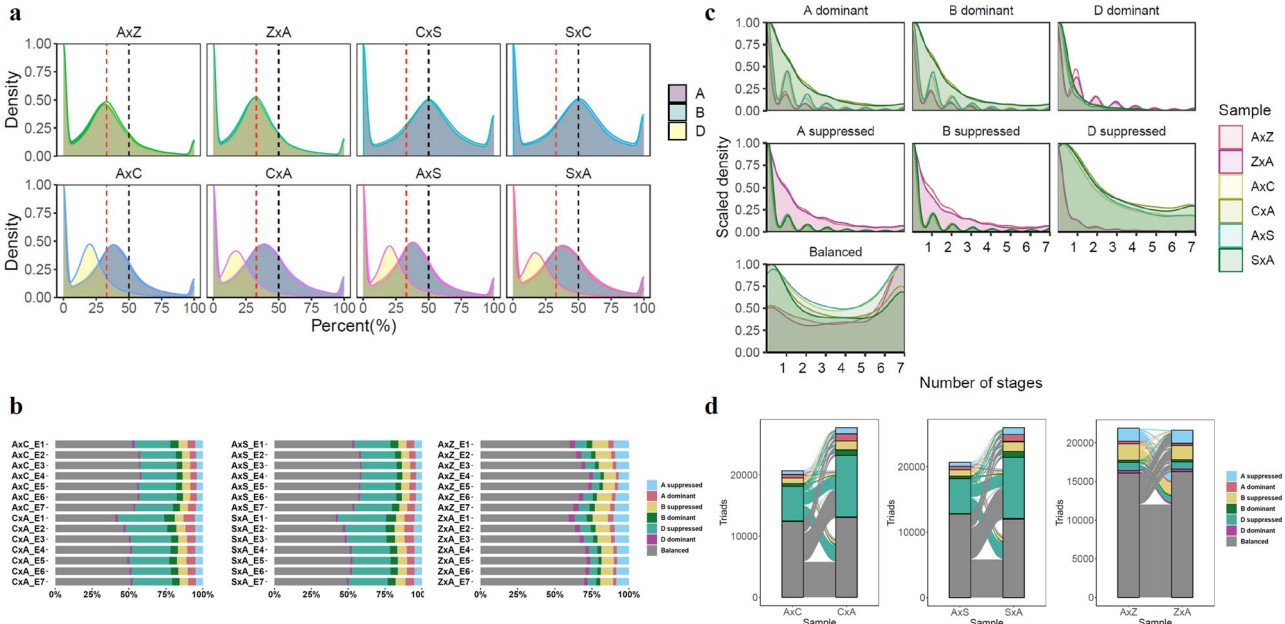

**Fig. 5 Homoeologous gene expression pattern after hybridization between polyploidy wheat species. a** Percent density of homoeologous triads on A, B and D genomes. **b** Percentage of homoeologue triads categorized into genome expression bias categories in reciprocal pairs at embryo stages. **c** Variation of triad expression patterns across seven embryo stages. Homoeologue triads were classified into the seven defined categories including balanced, A dominant, B dominant, D dominant, A suppressed, B suppressed and D suppressed, based on relative normalized expression within each triad as described by Ramirez-Gonzalez et al. (2018). **d** Alluvial diagrams depicting the shifts of homoeolog expression bias categories between reciprocal crosses. Links between samples represent the triads that flow to the same or a different category between reciprocal cross. Triads belonging to each of the seven categories were colored to show shifts of triads between different categories.

univalent D genome in pentaploids, with attenuated suppression in crosses using a higher ploidy female parent.

Polyploids may confer phenotypic plasticity by allowing homoeologs to be expressed differently across tissues and/or environmental conditions[15,37]. We further investigated the number of expression categories for each triad across embryo stages (Supplementary Fig. 23 and Fig. 5c). In general, balanced triads increased at later developmental stages and reached the highest number at stage seven. Other expression categories showed decreased numbers of triads at later stages (Fig. 5c). These results suggest balanced triads were relatively stable while dominant and suppressed triads tended to be more variable during embryo development. Hybridization between hexaploids (A×Z and Z×A) demonstrated similar distribution patterns while reciprocal crosses between tetraploids and hexaploids showed distinct distributions (Fig. 5c). Hybridization between tetraploids (AABB) and hexaploids (AABBDD) yielded pentaploids (AABBD) with a univalent D genome; these pentaploids contained a large number of suppressed D triads. In addition, the univalent D subgenome affected A and B subgenomes. More A dominant and B dominant triads but fewer A suppressed and B suppressed triads were found in pentaploids compared to hexaploids (Fig. 5c and Supplementary Fig. 24). Despite genetically identical embryos, differences were observed between the reciprocal crosses in pentaploids. Specifically, more A and B dominant triads were found in pentaploids in which the female parent was a tetraploid (C×A and S×A) than in pentaploids with a hexaploid female parent (A×C and A×S), while the D genome showed a similar distribution pattern (Fig. 5c).

To examine dynamic triads across embryo stages between reciprocal crosses, we identified switches of triads between categories (Supplementary Data 12). The changes in bias classification for each triad were traced between samples (Fig. 5d, Supplementary Fig. 25, and Supplementary Fig. S26). Triads most often remained consistent in their homoeolog expression bias

classification. Indeed, more than 40% of triads were retained in their categories after hybridization. More than 70% of triads were balanced in A×Z and Z×A (Fig. 5d). Unlike A×Z and Z×A, similar patterns of switches between reciprocal crosses were observed between pentaploids (A×C and C×A, A×S and S×A). Notably, more D suppressed triads were found in pentaploids compared to hexaploids. In addition, more D suppressed triads were found in pentaploids with a female tetraploid parent compared to pentaploids with female hexaploid parent (Fig. 5d). Specifically, a total of 1015 and 980 triads were switched from balanced to D suppressed between A×C and C×A as well as A×S and S×A, respectively, while only 135 triads switched to D suppressed between A×Z and Z×A (Supplementary Fig. 26). Similar biological processes were highly enriched in these genes amongst pentaploids including protein folding, DNA replication, mRNA splicing, and intracellular transport (Supplementary Data 13). In addition to differential homoeologous expression of the D genome, biased triad changes were also observed in the A and B subgenome, e.g., more A and B dominant triads were found in pentaploids compared to hexaploids. These results suggested that hybridization between species with different ploidy levels resulted in reprogramming of transcriptional regulation and an imbalance of homoeolog gene expression associated with A, B, and D genomes in embryos.

Endosperm contain two sets of genetic materials from the maternal genome and one set from paternal genome, meaning hybridization between hexaploids (A×Z and Z×A) produces an AAAABBBBDDDD endosperm genome. Although embryonic genetic materials are identical, the endosperm in pentaploids resulting from crosses using a hexaploid maternal parent (A×C and A×S, AAAABBBBDD) has two extra sets of the D subgenome than the endosperm of pentaploids created with a tetraploid maternal parent (C×A and S×A, AAAABBBD). Unlike embryos where a similar number of D suppressed triads were identified between reciprocal crosses, a large proportion of the D suppressed category

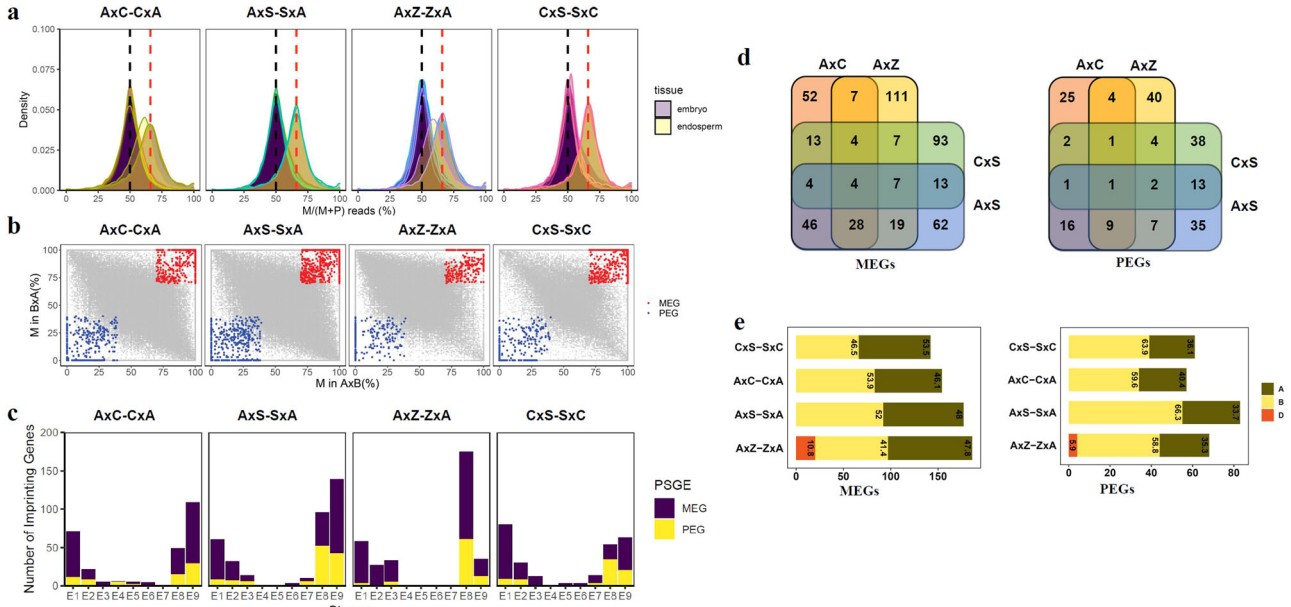

**Fig. 6 Allele-specific expression analysis in reciprocal crosses. a** Distribution of the percentage of maternal alleles in each reciprocal cross pair in embryo and endosperm tissue. Maternal fraction of reads (x-axis) are presented as percent of reads from maternal parent against read counts of the paternally and maternally derived reads in the reciprocal crosses, respectively. Black dash line represents 50% expected percent of maternal reads in embryos and red dash line denotes 67% expected percent of maternal reads in endosperm. **b** Allele-specific expression analysis in each reciprocal cross pair. x-axis (percent of M in A×B) represents maternal fraction of reads for genes from A×C, A×S, A×Z, or C×S while y-axis represents maternal fraction of reads for genes (percent of M in B×A) from C×A, S×A, Z×A, and S×C, respectively. **c** Number of maternally expressed genes (MEGs) and paternally expressed genes (PEGs) identified at each stage in each cross pair. For embryos, MEGs or PEGs were identified as more than 70% of total SNP-containing reads that were maternally or paternally derived (FDR < 0.01). For endosperm, MEGs were identified as more than 85% of total SNP-containing reads that were maternally derived while PEGs were identified as more than 60% of total SNP-containing reads that were paternally derived (FDR < 0.01). **d** Venn diagrams of imprinted genes identified from A×C and C×A, A×Z and Z×A, A×S and S×A, and C×S and S×C reciprocal cross pairs. **e** Distribution of imprinted genes on A, B, and D genomes in reciprocal crosses.

was found in endosperms of C×A and S×A compared to A×C and A×S (Supplementary Fig. 25 and Supplementary Data 12). These results suggest independent suppression of genes on the D subgenome in the embryo and endosperm of pentaploids.

**Genomic imprinting patterns during embryogenesis in polyploid wheats.** Parent-specific gene expression (PSGE) occurs when alleles inherited from the mother or father are expressed at unequal levels in the offspring. Genomic imprinting is a special case of PSGE in which an epigenetic chromosomal mark leads to complete silencing of one of the parental alleles[38]. To investigate parent-of-origin effects including maternally and paternally biased genes expressed during embryogenesis, we performed allele-specific gene expression (ASGE) analysis and generated lists of genes showing parent-of-origin biases, using 119307, 125469, 116448, and 54009 SNPs that distinguish maternal and paternal alleles in four pairs of reciprocal crosses (Supplementary Data 14).

Density plots of maternal reads vs. total reads showed an ~2:1 ratio of maternal reads vs. paternal reads in endosperm tissues and 1:1 ratio in embryo tissues in reciprocal crosses (Fig. 6a). Maternally and paternally biased genes between reciprocal crosses were identified, requiring that (a) the parental bias be in the same direction in each cross and (b) the ratio of maternal or parental reads to the total was greater than 0.7 for embryo stages. For endosperm stages, the ratio was 0.85 for maternal reads and 0.6 for paternal reads. A total of 658 imprinted genes were identified across all samples (Chi square test, FDR < 0.01) (Fig. 6b, Supplementary Fig. 27 and Supplementary Data 15); with 312 from the seven embryo stages and 374 from the two endosperm stages examined (Supplementary Data 15). The embryo and endosperm are genetically identical except for their ploidy level,

with the embryo having one maternal and one paternal dose and the endosperm having two maternal doses and one paternal dose. In different crosses, we observed 63–141 maternally expressed genes (MEGs) and 44–94 paternally expressed genes (PEGs) in endosperm (E8 and E9, Fig. 6c). In different crosses, 52–71 MEGs were identified at the E1 stage, followed by 14–26 in the E2 stage in embryos. A smaller number of PEGs (n = 1–11) were found in embryo tissues at the E1 and E2 stages (Fig. 6c). Both MEGs and PEGs decreased drastically at the E3 stage. Clustering analysis showed different MEGs and PEGs at each stage amongst tetraploids, pentaploids, and hexaploids, indicating that imprinted genes are not only regulated in a stage- and tissue-specific manner but also affected by ploidy levels (Supplementary Fig. 28).

Comparing imprinted genes between reciprocal cross pairs identified 109 overlapping MEGs and PEGs between reciprocal pairs of pentaploids (A×C-C×A, A×S-S×A) (Fig. 6d). Reciprocal pairs with hexaploids (A×Z-Z×A) showed 58 and 77 overlapping genes with A×C-C×A and A×S-S×A, respectively. Reciprocal pairs with tetraploids (C×S-S×C) had 30, 45, and 30 imprinted genes commonly found in A×C-C×A, A×S-S×A, and A×Z-Z×A, respectively (Fig. 6d). Hence, reciprocal crosses with the same ploidy level share more imprinting genes. Distribution of MEGs and PEGs in A, B, and D subgenomes were also examined. Owing to the univalent D genome in pentaploids and the lack of a D genome in tetraploids, MEGs and PEGs were identified in the A and B subgenomes in pentaploids and tetraploids. In reciprocal crosses between hexaploids (A×Z-Z×A), about 41.4% of MEGs and 58.8% of PEGs were from the B genome, 47.8% of MEGs and 35.3% of PEGs were from the A genome, and only 10.8% of MEGs and 5.9% of PEGs were from the D genome. These results show a biased expression of imprinted genes from the A, B, and D genomes (Fig. 6e).

**Validation of DEGs, AS events and imprinting genes using digital PCR (ddPCR).** To validate our results from DEG, AS and imprinting gene analyses, we selected several genes and transcripts for droplet digital PCR (ddPCR) assays using specifically designed primers and probes. For the validation of DEGs, gene-specific primers were designed for the target genes to assess gene expression changes in different stages. The expression levels of three DEGs, *XLOC_061322* on chr2D, *XLOC_123600* on chr4D and *XLOC_114665* on chr4B in embryos at stage E4, E5 and E6 from the cross between Strong Field and AC Barrie (S×A) were quantified and compared (Fig. 7a). Both RNAseq and ddPCR showed similar log2Fold-Change values between E5 and E4 as well as E6 and E4, suggesting good correlation between the two techniques (Pearson correlation value = 0.976).

For the validation of AS events, primers targeting one or multiple different transcripts were designed and the expression of each transcript was calculated. AS site-spanning primers were designed with two fluorescent probes and 2–3 base pair mismatches to ensure accurate detection and quantification of transcript isoforms from each homoeologous gene (see details in Methods). We examined the expression level of four AS events at stage E5 and E6 in the reciprocal cross pair of S×A and A×S using ddPCR. Our results showed a high correlation between RNAseq and ddPCR (Fig. 7b, Pearson correlation value = 0.839).

Selected imprinted genes between S×A and A×S were confirmed in embryo and endosperm. Two genes, *XLOC_045394* on chr2B and XLOC_116226 on chr4B, were imprinting genes from embryos at E3 stage. *XLOC_193638* on chr7A and *XLOC_210220* on chr7B were identified as imprinting genes from endosperm at E8 stage. Allele-specific fluorescent probes were designed based on the SNPs identified between two parents of S×A and A×S, respectively. Droplets from 5'-labeled with 6-carboxyfluorescein (FAM) and 6-carboxy-2, 4, 4, 5, 7, 7 hexachlorofluorescein succinimidyl ester (HEX) probes were clustered in the 2-D droplet amplitude plots and the ratio of FAM and HEX droplets were calculated to compare the ratios of maternal or parental reads from RNASeq (Fig. 7c, d). The results showed that the ratios of maternal reads to total reads from RNAseq were consistent with those from ddPCR droplets to the total droplets in tested genes at embryos in E3 and endosperm in E8 (Fig. 7e). Thus, the expression pattern of selected MEGs and PEGs was independently validated. Together, our results indicated an overall reliability of the differential expression analysis, AS events and imprinting gene identification.

## Discussion

The dynamics of transcription in seed development have been explored extensively in plants[26,30,39,40], but the effects of interspecific hybridization on gene expression and regulation networks, particularly when merging two polyploid genomes were unclear. In this study, we analyzed transcriptional changes throughout embryo development in the progeny from reciprocal crosses between hexaploid and tetraploid wheat species. A consensus transcriptome was constructed to investigate transcriptional dynamics across multiple samples with different ploidy levels. A correlation matrix and phylogenetic tree produced by comparing expression levels across all genes between reciprocal samples demonstrated clear separation of samples from different tissues, developmental stages and ploidy levels. Two sets of differentially expressed genes (DEGs) during embryo development were identified: (1) stage-DEG, from comparisons between each time point against the two-cell stage for each sample, and (2) sample-DEG, from comparisons between reciprocal crosses at the same stage.

Many stage-DEGs were found in reciprocal crosses and the majority were identified in F1 hybrids regardless of whether reciprocal crosses were made between species of the same or different ploidy levels. Comparing gene expression patterns across samples with different ploidy levels and varying numbers of chromosomes remains a major challenge especially for large-scale analyses[23]. Conserved gene expression patterns were observed between different pairs of reciprocal crosses, indicating that differences between developmental stages are primarily responsible for the transcriptional changes. Clusters with similar expression patterns tended to have similar biological functions. This result suggested that during embryogenesis, genes on the univalent D genome in pentaploids are actively expressed, and are likely to have similar expression patterns as the bivalent D genome in hexaploids.

To investigate cross-directional effects in reciprocal crosses, we identified sample-DEGs in reciprocal cross pairs. Reciprocal pairs of pentaploids showed twice the number of DEGs relative to hexaploid or tetraploid pairs. This observation suggests that hybridization between different ploidy levels yields active transcript changes. It has been suggested that hybridization between species with different ploidy levels leads to major genomic stress and reprogramming of gene expression[18]. The extensive transcriptional changes observed in reciprocal crosses between hexaploid and tetraploid, therefore, may indicate genomic stress. Investigation of biological pathways revealed an enhanced protein synthesis, active AS and chromatin remodeling in pentaploids whose female parent is hexaploid, suggesting post-transcriptional and epigenetic regulations on gene expression in reciprocal crosses[12,13].

AS is an important post-transcriptional mechanism that regulates gene expression[31,41]. Changes in AS patterns have been observed in polypoid plants during genome duplication[12,42], while little is known about AS changes during embryogenesis after hybridization between polyploid species with different ploidy levels. Consistent with previous studies[30], AS events increased as embryonic development progressed (Fig. 4a). Fewer AS events were identified in the D subgenome in both hexaploids and pentaploids than the A and B subgenomes (Fig. 4b), which might be related to the late incorporation of the D genome during evolution[13].

Interspecific hybridization between two plant species with different ploidy levels usually employs the higher ploidy level species as the maternal parent to support higher seed set and germination[5]. Consistent with an up-regulation of genes involved in mRNA splicing in pentaploids with a hexaploid female parent, a higher level of AS was found in pentaploids with a hexaploid mother than in pentaploids with a tetraploid mother. The uniparental AS patterns observed in reciprocal-cross hybrids indicate that parental genomes might not contribute equally to the F1 hybrid genome. It is possible that the higher ploidy species as maternal parent has diverged AS patterns, which might support regulatory flexibility in duplicated genes, in turn resulting in a higher fertility rate. In addition, AS allows a single gene to generate multiple transcripts to enhance transcriptome plasticity and proteome diversity[12,30]. The extensive changes in AS after interspecific hybridization between polyploids may serve as an important mechanism to alleviate transcriptome shock or genomic stress.

Variations in gene expression, including differential homoeolog expression and homoeolog silencing (non-expression of one homoeolog) have been observed in allopolyploid species[37,43–45]. These changes could be derived from hybridization and polyploidization, which lead to biased homoeolog expression[39,43]. Pentaploid wheat hybrids show the predominance of heterozygous loci in their A and B subgenomes and haploid D subgenome. In embryos, we observed a balanced expression pattern of homoeologous genes in the A, B and D genomes in hexaploids or A and B genomes in tetraploids, while an imbalanced expression pattern was observed in pentaploids. Specifically, the contribution of homoeologous genes on the univalent D genome to total triad

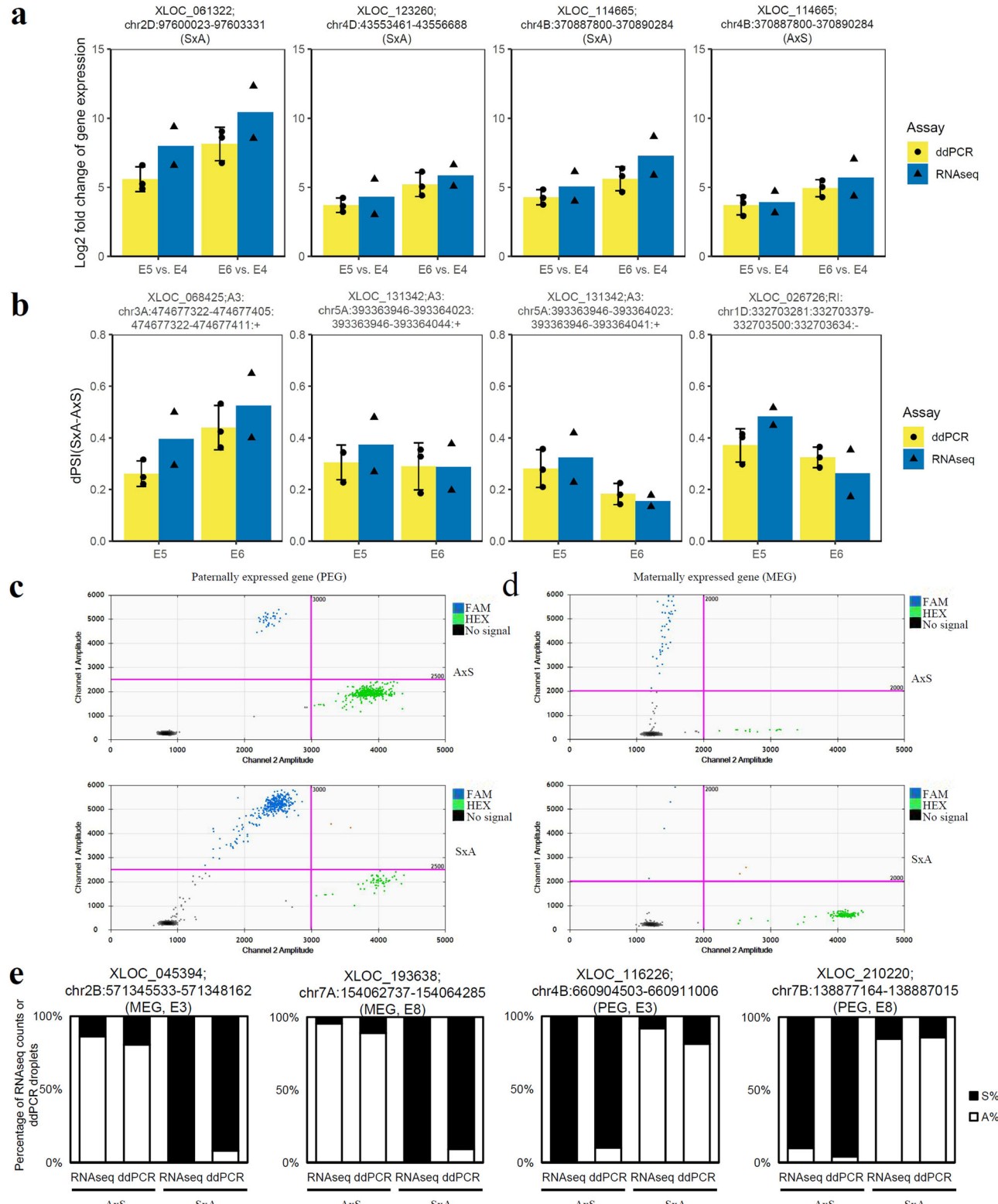

expression was repressed in pentaploids. Homoeologs from the A and B subgenomes were shifted to a higher proportion in pentaploids. Therefore, hybridization between wheat species with different ploidy levels not only repressed gene expression on the univalent D genome, but also appeared to interfere with homoeolog expression on A and B genome.

Homoeologous triads most often remained consistent in their homoeolog expression bias classification, a phenomenon seen across various tissues in hexaploid wheat[44]. In this study, balanced triads remained balanced across all seven stages of embryo development (Supplementary Fig. 24). Many balanced triads in pentaploids with a hexaploid female parent (A×C and A×S) were categorized as D suppressed in C×A and S×A in which the female parent is tetraploid. On the contrary, a number of D suppressed triads in A×C and A×S were classified as balanced in C×A and S×A. In interploidy hybridizations, proper embryo development depends

**Fig. 7 Validation of DEG, AS and imprinting analysis results through droplet digital PCR (ddPCR) assay. a** Expression of select DEGs assayed by RNAseq (blue) and ddPCR (yellow) (*n* = 2–3) in different stages and different hybridizations. Chromosome number, genome position and crosses are shown below gene name. *x*-axis, comparison categories; *y*-axis, log2fold-change of gene expressions. Values represent the mean and error bars depict the standard deviation. **b** dPSI of select AS events assayed by RNAseq (blue) and ddPCR (yellow) (*n* = 2–3) in different stages and different hybridizations. dPSI was calculated based on the PSI of select AS events in S×A and A×S lines. PSI was calculated based on the gene expression of all transcripts involved in the corresponding AS events. Chromosome number, AS type, genome position information are shown after the gene name. *x*-axis, stages; *y*-axis, dPSI. Values represent the mean and error bars depict the standard deviation. **c, d** 2-D droplet amplitude plots of ddPCR assays (*n* = 2–3) for imprinting analysis. The ddPCR probes for imprinting analysis were designed based on the SNPs between the two lines used to generate the hybridization lines, and then were synthesized with FAM and HEX fusion, respectively. Examples for a paternally expressed gene (PEG) and maternally expressed gene (MEG) were displayed, showing the significantly different ratio of FAM and HEX associated signals. *x*-axis, the amplitude of droplets in HEX channel; *y*-axis, the amplitude of droplets in FAM channel. Blue dots, droplets with FAM signals; green dots, droplets with HEX signals; black dots, droplets without fluorescence signal. **e** Ratios of maternal or parental reads/droplets of select imprinted genes assayed by RNAseq and ddPCR (*n* = 2–3) in different stages and different hybridizations. Ratios of maternal or parental reads to the total from RNAseq were calculated based on our pipeline described in Methods, and ratios of maternal or parental droplets to the total were calculated based on the associated FAM and HEX droplets identified from the separation of 2-D droplet amplitude plots like shown in c and d. From left to right and representative MEG in embryo at E3 stage, MEG in endosperm at E8 stage, PEG in embryo at E3 stage, PEG in endosperm at E8 stage, respectively. Chromosome number, genome position, MEG or PEG and stages are shown after the gene name. *x*-axis, different assays applied in different stages; *y*-axis, percentage of RNAseq reads or ddPCR droplets. No significant difference (Student *t*-test) was identified between the results from RNAseq and ddPCR.

---

on the ploidy ratio of the maternal and parental parent[46]. The ploidy level of the embryo and its associated endosperm appears critical for embryogenesis[3,5]. In pentaploids, although the genetic components are identical in embryos, the endosperm of pentaploids with hexaploid as maternal parent (AAABBBDD) has two extra sets of D genome compared to which using tetraploid as maternal parent (AAABBBD). A much higher number of D suppressed triads were observed in C×A and S×A, relative to A×C and A×S (Supplementary Fig. 25). It is possible that changes in triad classification between reciprocal crosses with different polyploidy levels was caused by the cytoplasmic-nuclear interactions[46]. The disruption of balanced gene expression, especially on D genome, in the endosperm of pentaploids with tetraploid female parents might affect the fertility of its progeny.

A large number of imprinted genes have been identified in the endosperm of various plant species[39,47–51]. Likewise, our study identified hundreds of maternally and paternally imprinted genes in wheat endosperm. Previously, only a few or tens of imprinted genes in embryos had been described in maize, Arabidopsis and rice[52–56]. We identified 213 maternally expressed genes (MEGs) and 37 paternally expressed genes (PEGs) in wheat embryos at E1 and E2 stages, with MEGs far outnumbering PEGs. The number of MEGs decreased rapidly after the E2 stage. Whether these several hundred genes that show predominantly maternal parent-of-origin bias truly reflect imprinting or the last stage of delayed paternal genome activation is not clear[57]. Maternally biased gene expression during zygotic genome activation has been shown to affect thousands of genes for a short period after fertilization[56,58]. Thus, the hundreds of MEGs identified in our study (rather than thousands) are more reminiscent of imprinting. Our results suggest that genomic imprinting occurs in early embryogenesis in wheat at a larger scale than previously described in other plant species.

Though many imprinted genes have been identified and the epigenetic mechanisms that regulate them are widely studied, little is known about genomic imprinting after hybridization between polyploids with different ploidy levels. Different sets of genes can be imprinted at different stages of seed development[59]. A previous study identified a total of 283 imprinted genes in developing endosperm of diploid, tetraploid and hexaploid wheat[39], while we detected 658 from embryos and endosperm of all four cross pairs in this study. More imprinted genes were identified from embryos in this study and there were 62 genes commonly identified in endosperm (Supplementary Data 15). The discrepancy may be due to different tissue types and development stages collected in this study. Indeed, we showed that imprinted genes are regulated in a stage- and tissue-specific manner. Owing to the differences in genome organization and epigenetic modification, only a few imprinted genes were commonly identified in all three ploidy levels (tetraploids, hexaploids, and pentaploids). However, two sets of pentaploids share more imprinted genes, suggesting ploidy-dependent expression of imprinted genes. This is perhaps not surprising given that each stage of grain development has its own transcriptome and reveals that genome-wide imprinting occurs at multiple tissue and developmental stages.

## Methods

**Plant material, growth, and isolation of developing embryo and endosperm tissues.** Polyploid (A, AC Barrie; Z, Chinese Spring) and tetraploid (C, Commander; S, Strong Field) wheat plants were grown in growth chambers under long-day conditions of 16 h of light, 22 °C and 8 h of dark, 20 °C, with light intensity of 100–120 μmol m$^{-2}$ s$^{-1}$ (Philips high-output F54T5/835-841 bulbs). Spikelets were emasculated and pollinated at the heading stage to ensure developmentally coordinated F1 seed production for embryo and endosperm isolation[26]. The hours/days after pollination and the morphologies of developing embryos were used for classifying the developmental stages of the embryo and endosperm (Supplementary Data 1). The embryo and endosperm come from different developmental stages of the same seed, and then were separated under a microscope using tweezers at defined stages. For each embryo sample in early stages of development, about 30 embryos were pooled in each biologically replicated sample. For each sample in late embryo stages, a minimum of 10 embryos were pooled in each biological replicate sample. Embryos were carefully washed twice with isolation solution exchanges to ensure a clean collection of embryos. Embryo stages were visually identified and confirmed with a compound microscope (Leica DMR) and images were taken using a MicroFire camera (Optronics). A minimum of 10 grains were used for endosperm isolation in each biological replicate sample. Representative endosperm samples was stained with Toluidine blue and identified using light micrographs.

**RNA-sequencing data processing and mapping.** Total RNA was extracted from embryo and endosperm tissues in different developmental stages of reciprocal crosses (A×C, C×A, A×S, C×S, S×A, A×Z, and Z×A), following the protocol for the RNAqueous-Micro kit (Ambion, Catalog# 1927). Two biological replicates for each sample were collected. RNA-seq libraries were prepared according to Illumina's instructions, using the TruSeqRNA Sample Preparation Kit v2 (Illumina) and pair-end sequencing (126 cycles) was conducted on Illumina HiSeq 2500 following the manufacturer's protocols. Quality check of the raw RNA-Seq data were processed by FastQC v0.11.9[60]. The adapter removal and quality trimming were carried out using Trimmomatic v0.38 with default parameters[61]. The filtered reads were mapped to IWGSC Version 1.1 survey genome assembly using STAR-2.4.2a[62]. Specifically, for hexaploids (AABBDD) and pentaploids (AABBD), the raw RNA-seq sequence reads were aligned to the entire IWGSC RefSeq 1.1 genome assembly and the tetraploid samples (AABB) were aligned to the A and B subgenomes.

Different assembly methods were performed to annotate high-fidelity gene models based on multi-sample RNA-seq data. To get a more comprehensive and functional annotation, the following tools were used. The mapped BAM file from STAR and reference annotation IWGSCv1.1 HC were provided as input to StringTie v2.1.4 to assemble transcripts for each sample[63]. The genome-guided

transcript assemblies were then served as input to Cuffmerge[64], TACO[65] or TransDecoder (http://transdecoder.github.io) for transcript identification. StringTie was used for generating reference-guide annotation file and new gene models were predicted by Cuffmerge or TACO[64]. TransDecoder was used for new gene annotation further filtered by finding coding regions within transcripts by using Cuffmerge or TACO predicted cDNA sequence as input[66]. *TransDecoder.LongOrfs* and *TransDecoder.Predict* was used to extract the long open reading frames and predict the likely coding regions, then the *cdna_alignment_orf_to_genome_orf.pl* in TransDecoder package was used to generate the final annotation file with the filtered gene model. The accuracy of nine different assembled annotations was assessed for the sensitivity and precision of transcript identification by comparing them to the reference using *gffcompare* 0.12.1. *featureCounts*[67] was then employed to count the reads spread across the exonic regions of each gene model.

**Gene functional annotations**. A total of 133,364 high-fidelity gene models and the longest protein sequence was extracted for each gene used as input in the following annotation step. These predicted high-quality genes were annotated with Gene Ontology (GO) and Kyoto Encyclopedia of Genes and Genomes (KEGG) by using InterProScan v5.40–77.0[68] in local with parameters -goterms -iprlookup -pa. In parallel, known orthologs in *Arabidopsis thaliana* and *Oryza sativa* (*ssp*. japonica) were sought from EnsemblePlants v28, identified by local blastp (v2.9.0) with a threshold criteria of $E \leq 10\text{-}6$, minimum alignment (peptide) length ≥50, Similarity ≥50%, and bit score ≥50. The custom functional GO annotation was used to calculate GO enrichment for gene clusters via Fisher's exact test and was performed using the *clusterProfiler* R package[69]. Fisher's exact tests with FDR < 0.05, computed by taking the whole transcriptome as background, were considered significantly enriched. MapMan bins of wheat genes were assigned by the Mercator pipeline for automated sequence annotation (http://www.plabipd.de/portal/mercator-sequence-annotation)[70]. A value of 50 was used as BLAST_CUTOFF. Genes involved in AS pathways were compiled from SRGD datase (http://www.plantgdb.org/SRGD/)[32]. The list of embryo essential genes was obtained from Xiang et al.[26].

**Differential expression analysis and gene coexpression analysis**. Differential expression analysis was performed using DESeq2 to identify stage-DEGs and sample-DEGs[71]. For stage-DEGs, comparisons between each time point against the two-cell stage at embryo stages for each sample were calculated. For sample-DEGs, read counts from each pair of reciprocal crosses were normalized, and comparisons were calculated between F1 embryos at each stage. Gene expression was assessed by transcript per million (TPM), which is quantified on a concentration basis by dividing read abundance by a factor of the whole library. DEGs were identified for each comparison ($|\log2FC| \geq 1$, FDR ≤ 0.01 and greater than one TPM across all samples) and p-values were corrected by Benjamini–Hochberg [FDR] method[72]. An ANOVA test was further applied to identify genes with different expression levels between reciprocal crosses across all stages.

Normalized counts were transformed into variance-stabilizing values using DESeq2[71] and clustering was performed using WGCNA[73] with a soft thresholding power of nine. Closely connected genes were identified by hierarchical clustering based on the topological overlap matrix and cutting the resulting dendrogram with the dynamicTreeCut program with deepslipt = 2, minModuleSize = 30 to obtain dynamic gene clusters. Initial clusters with similar expression profiles were merged at cutheight = 0.25. Module eigengene (ME) values that represented the first principal component of each cluster were calculated. GO enrichment was performed for each cluster using the *clusterProfiler* R package[69]. Significant GO terms in biological processes were slimmed by grouping similar terms based on their semantic similarity using customized *rrvgo* package[74].

**Homoeologs identity and analysis of expression dominance**. Homoeologs were identified by MCScanX[75,76]. Briefly, homoeologous genes in A, B, and D subgenome were identified using the longest transcripts seq by pair-wise synteny search using *jcvi.compara.catalog ortholog* command. Homoeolog triads were identified by the *jcvi.compara.synteny mcscan* and *jcvi.formats.base join* commands with default parameters. Homoeologue triads were classified into the seven defined categories (balanced, A dominant, B dominant, D dominant, A suppressed, B suppressed, and D suppressed) based on relative normalized expression within each triad. The definition of homoeolog expression patterns was as previously described[44]. Assuming that the expression of homoeologous genes from different subgenomes is 1, homoeolog expression patterns will be classified as: Balanced = c(0.33,0.33,0.33), A dominant = c(1,0,0), B dominant = c(0,1,0), D dominant = c(0,0,1), A suppressed = c(0,0.5,0.5), B suppressed = c(0.5,0,0.5) and D suppressed = c(0.5,0.5,0). The Euclidean distances between each gene and each classification were calculated based on the fraction of the reads mapped to the given gene triad, and the minimum distance was considered to be the expression pattern of triad.

**Analysis of AS events**. SUPPPA2 was used to generate a set of AS events from the annotated isoforms in the IWGSC ref1.1 genome, using the *generateEvents* mode to detect retained introns, skipped exons, and alternative first or last exons, and mutually exclusive exons (-e SE MX RI SS FL), with 10 bp as the minimum exon length (−l 10)[29]. We also calculated the TPM values at the isoform level using Salmon 0.10.2[77]. SUPPA2 *psiPerIsoform* mode was then used to calculate the inclusion rates of each isoform (PSI: percentage spliced-in) in each embryo sample, using the expression levels of each isoform (obtained from Salmon) as a reference. For samples of different ploidy levels, the reference used in Salmon was the same as the RNA-seq SATR mapping step. Differential splicing was quantified by calculating the average difference in PSI values between each sample group, and p-values were obtained using the empirical significance calculation method described in SUPPA2[29].

**SNP detection and allele-specific transcript analysis**. Raw RNA-sequencing data of reciprocal crosses generated in this study and raw reads of the four parents collected at the sample developmental stages (A, AC Barrie; C, Commander; S, Strong Field; Z, Chinese Spring)[26,30] were used for SNP and allele-specific gene expression analysis. According to the pipeline for imprinting analyses described by Picard and Gehring[78], raw reads were quality filtered and aligned to the IWGSC1.1 reference genome for SNP identification between reciprocal crosses using STAR[62]. SNP detection and identification of SNP-associated read raw sequencing data from each library were preprocessed to filter out clipped adapter sequences, contaminated sequences, low-quality reads and reads that mapped to more than one position in the reference genome. Alignments of two replicates for each sample were merged for SNP Calling by Sentieon[79] using the Haplotyper algorithm. All possible SNPs were detected by merging a total of 216 sample gvcf files to one vcf file. To reduce the detection rate of false-positive SNPs, SNP detection used the following stringent criteria: (i) the sufficient base quality (Q-values > 20); (ii) at least ten reads coverage across SNP site; and (iii) exclusion of heterozygous sites of each sample. The remaining SNPs between reciprocal crosses defined as high-quality were used for subsequent analysis (Supplementary Data 14).

Allele-specific SNPs between the two parents were identified and subsequently used for SNPs identification in F1 hybrids in vcf files. The paternal SNPs located in the gene were processed into allele-specific, per-gene counts using the identified SNPs between male and female parents. Paternally derived in two reciprocal crosses of both biological replicates (with a minimum of 10 SNP-associated reads per cross) were identified as imprinted genes. Genes with $\chi^2$ goodness-of-fit test (FDR < 0.01) and ≥70% of total SNP-containing reads that were maternally or paternally derived were identified as maternally or paternally bias imprinted genes in embryos. For endosperm, the ratio is 0.85 for maternal reads and 0.6 for parental reads (FDR < 0.01).

**Validation of RNAseq datasets using a droplet digital PCR (ddPCR) assay**. The mRNA was reverse transcribed following the SuperScript IV VILO system (Thermo Fisher Scientific, Cat#: 11756050) according to the manufacturer's instructions. Transcript abundance was measured using the Bio-Rad QX200 ddPCR System (Bio-Rad). The ddPCR EvaGreen Supermix (Bio-Rad) was used for DEG and AS validations and ddPCR SuperMix for probes reaction mixture (no dUTPs, Bio-Rad) was used for imprinting validations. In brief, each 20 μL 1x ddPCR SuperMix reaction mixture containing cDNA templates, forward and reverse primers (and two probes) with optimized concentration were mixed with 70 μL of Droplet Generation oil for Probes (Bio-Rad) in a DG8 Cartridge (Bio-Rad). Probes were 5'-labeled with 6-carboxyfluorescein (FAM) or 6-carboxy-2, 4, 4, 5, 7, 7 hexachlorofluorescein succinimidyl ester (HEX) as the reporter and ZEN/Iowa Black FQ as the 3'-labeled double quenchers. The cartridge was covered with a DG8 gasket and loaded into the QX200 Droplet Generator (Bio-Rad) to generate PCR droplets. From each droplet mix, 40 μL was transferred to a 96-well PCR plate (Bio-Rad) and sealed using PX1™ PCR plate Sealer (Bio-Rad). PCR thermal cycling was optimized, and the amplification signals were read using the QX200™ Droplet Reader and analyzed using QuantaSoft software (Bio-Rad) in 2-D mode. Three biologically independent samples (n = 3) were used for each validation. For primer design, crossing introns were considered to reduce the genomic DNA contaminations. The detailed primers and probes information can be found in Supplementary Data 16.

**Statistics and reproducibility**. Details about the statistical analyses used in this study are described in the corresponding methods section. Differential expressed genes were identified using log2Fold-Change ($|\log2FC| \geq 1$) and p-values were corrected by Benjamini–Hochberg [FDR] method (FDR < 0.01). ANOVA test (FDR < 0.05) was performed for each gene to identify differentially expressed genes between reciprocal crosses across all stages. For Significantly enriched GO terms were calculated using Fisher's exact test (FDR < 0.05). The distribution of PSI between reciprocal crosses with parents from different ploidy levels was measured using Kolmogorov–Smirnov test. DAS values were identified through pair-wise comparisons between reciprocal crosses ($|\Delta PSI| > 0.1$ and FDR < 0.05). Imprinting genes were assessed with $\chi^2$ goodness-of-fit test (FDR < 0.01). In all these cases, p-values were corrected using the Benjamini–Hochberg [FDR] method. For ddPCR experiments, three biologically independent replicates were performed. Statistical significance of grain traits (each trait with at least three replicates, $n \geq 3$) between reciprocal crosses was performed using two-tailed Student t-test (p-value < 0.05).

**Reporting summary**. Further information on research design is available in the Nature Portfolio Reporting Summary linked to this article.

## Data availability

All RNA-seq raw data generated from this study can be found in the Gene Expression Omnibus under accession number GSE188277. All the supplementary figures (Supplementary Figs. 1 to 28) and supporting datasets (Supplementary Data 1 to 16) are available in Supplementary Information. The source data underlying graphs are provided as a Supplementary Data 17. All other data and research materials are available by contacting the corresponding authors.

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

## Acknowledgements

We thank Dr. Wentao Zhang for reviewing the article and providing suggestions. The analysis of RNA-seq data was in part run on the bioinformatics computing platform of the National Key Laboratory of Crop Genetic Improvement, Huazhong Agricultural University. This work was supported by the Wheat Flagship Program of Aquatic and Crop Resource Development Research Division of the National Research Council of Canada (ACRD manuscript number 58293). The author responsible for the distribution of materials integral to the findings presented in this article in accordance with the policy described in the Instructions for Authors is: Daoquan Xiang (daoquan.xiang@nrc-cnrc.gc.ca).

## Author contributions

D.X., R.D., Q.L. conceived and coordinated the study; P.G., H.S., J.S, H.Y., and D.X. performed experiments; Q.L., Z.J., P.G., T.D.Q., C.S.G., and D.X. performed data analysis, prepared figures, and wrote the article; Z.J., F.Y., H.S., J.S., J. G., T.C., B.Y., Q.L., and Q.Y. contributed to bioinformatics data analysis, drafting the figures and tables; L.V.K., J.Z., N.P., C.S.G., and R.D. contributed to materials, reagents, and article preparation. All authors read and approved the final article.

## Competing interests

The authors declare no competing interests.
