## [Peer Review File · Communications Biology]

Reviewers' comments:

Reviewer #1 (Remarks to the Author):

The authors investigated genome-wide maternal and paternal contributions to polyploid grain development and analyzed transcriptomes from zygote to mature stage embryos as well as two endosperm stages derived from reciprocal crosses between tetraploid and hexaploid wheats. Results are novel and attractive, especially the homoeologous gene expression of D genome in the pentaploid and the identification of imprinted genes. However, some evidence needs to be provided to support the conclusion, and a comprehensive comparative analysis of the imprinted genes identified by the authors with previous results is required. Details are as follows:

Point 1:

In the introduction, the authors demonstrated the background and significance of this study. However, the introduction of background is not enough, for instance, Lines 60-61 "negative or additive effects" for details and hybridization barriers and sterility in polyploid wheat for details.

Point 2:

2.1 Identified imprinted genes: In Results, embryo-imprinted genes were defined with "the ratio of maternal or parental reads to the total was greater than 0.7 for embryo stage" were analyzed and identified imprinted genes (Lines 430-431). However, authors identified embryo-imprinted genes in the Experimental procedures by using "≥85% of total SNP-containing reads that were maternally derived or ≥70% that were paternally derived were identified as maternally bias imprinted genes in embryos" (Lines 738-740). Which ratio was used for identifying the embryo-imprinted genes?

2.2 Confirm the asymmetric of imprinted genes: In this study, a lot of imprinted genes were identified during embryogenesis in polyploid wheats. However, it is unclear that these imprinted genes showed allelic imbalance and asymmetric expression in fact. Please randomly select some candidate imprinted genes to confirm the asymmetric expression by using the RT-PCR sequencing or allele specific RT-PCR which is done for example in Chen et al., 2018.

2.3 Discussion the imprinted genes in polyploid wheat: The discussion of imprinted genes in this study is not depth. Yang et al (2018) suggested that genomic imprinting was evolutionarily conserved during wheat polyploidization by identified imprinted genes in diploid, tetraploid, and hexaploid wheat, respectively. Molecular evolution and preferential expression of imprinted genes can be discussed in this study by compared with the imprinted genes reported by Yang et al (2018), and these comparisons are favorable to improve the credibility of the gene imprinting status in this study.

Point 3:

About the plant material 'strongfield'. Lines 97/1023 showed "Strong Field", Lines 607/721 showed "strongfield". Please correct it.

Point 4:

Minor points:

Line 67—Please correct "Triticum turgidum ssp. Durum" into "Triticum turgidum ssp. durum".

Lines 84/248/508/667/703—abbreviation AS should in here.

Line 104—"two stages of endosperm (transition and leaf late stage) ", while in DataS1, two stages of endosperm are early and late.

Lines 521-523—"Comparisons between reciprocal crosses and their parents showed similar numbers of AS events between pentaploids and hexploids while tetraploids showed AS events". This sentence sounds somewhat contradictory.

Line 578—"maternally- and paternally imprinted genes", remove "-".

Lines 626/725—Please, correct "Dobin et al., 2012" into "Dobin et al., 2013".

Lines 635/638—Please, correct "Trapnell et al., 2012" into "Trapnell et al., 2010".

Line 669—Please, correct "were obtained from Xiang et al., 2019" into "were obtained from Xiang et al (2019)".

Lines 678/708—abbreviation TPM should in here.

Point 5:

Lines 719-722—"raw reads of the four parents collected at the sample developmental stages (A, AC Barrie; C, Commander; S, strongfield; Z, Chinese Spring) (Xiang et al., 2019) were used for SNP and

allele-specific gene expression analysis". However, 'Commander' and 'Chinese Spring' were not found in Xiang et al., 2019.

Point 6:

References 13/60. Authors' name should be corrected.

References 34/55 are mentioned in the References but are missing in the text.

Reviewer #2 (Remarks to the Author):

This study analyzed the differentially expressed genes, alternative splicing and parental allele expression patterns of embryos and endosperms in the reciprocal crosses of hexaploidy wheat, tetraploid wheat as well as pentaploid wheat, and the authors also compared homeologous gene expression patterns in different tissues of diverse samples. The study provides a large set of data for developing embryo and endosperms in wheat. However, the data presentation needs lots of improvement to fully support the conclusion. The following suggestions may help to improve the manuscript, which is encouraged to resubmit with revisions.

1. The title is not suitable for the manuscript.
2. The authors should provide phenotypic analysis of developing endosperm and embryo between interploidy reciprocal crosses, and it is better to analyze the association between gene expression and phenotypic variation.
3. The authors included too much information in the manuscript, e.g. DEGs, AS genes, homeologous gene variation and parental allele expression (imprinting). It might be better to focus on two aspects, and provide more detailed information.
4. The validation of imprinting, AS, DEG is needed, especially, previous data indicated that imprinted genes is mainly occurred in endosperm and very few were identified in embryo. The author identified a set of imprinted genes in wheat embryo, should provide confident evidence for this finding.
5. The authors should provide more detailed information in the methods section. 1) how do the authors analyze the reads in different biological replicates when identifying imprinted genes, the criteria of >5 SNP-associated reads is not sufficient; 2) when analyzing gene expression patterns of endosperm in pentaploid wheat, how do the author normalize the sequencing reads, since the reciprocal crosses have different sets of chromosomes; 3) The author declared two replicates of samples were collected, are they separately sequenced or mixed for sequencing? Should be stated clearly.
6. The writings needs improvement.

Reviewer #3 (Remarks to the Author):

Interspecific pentaploid F1 hybrids generated by crossing hexaploid and tetraploid wheat species have distinctive genetic variability due to chromosomal reconstitution which has great potential to improve several agronomic traits like disease resistance, abiotic stress tolerance, and grain quality. However, these interspecific pentaploid F1s suffer from sterility, reduced seed set, and pollen incompatibility, besides other deficiencies. In this manuscript by Jia et al., authors investigated gene expression in pentaploid F1s derived from crosses between two hexaploid wheat species - Chinese Spring and AC Barrie, and two tetraploid varieties - Strong Field and Commander. The authors analyzed seven stages from zygote to mature embryo during embryonic development and two endosperm stages. Reciprocal crosses were carried out to look into parental contributions to gene expression during these stages of seed development and observe the parent-of-origin-dependent gene expression as well.

First authors established expression profiles in different ploidy levels for different genome combinations (hexaploidy = AABBDD, pentaploid =AABBDD, and tetraploid =AABB) as a reference for investigating transcriptomes in their reciprocal crosses and developmental stages. As one can expect, gene expression in hexaploids with ABD genomes was closer to pentaploids (with the same ABD

genomes) than tetraploids with just AB genomes. Comparisons were made between reciprocal crosses and different samples. Analysis of DEGs showed an increase with temporal development; hexaploids and pentaploids have more DEGs compared to tetraploids, and almost 60% of DEGs were shared between each reciprocal cross pair.

Analysis from other studies has led to an interesting observation that using the higher ploidy species as the female parent can lead to reduced sterility in the wheat F1 pentaploids as the authors have correctly mentioned in the introduction. The authors show that the pentaploids with higher ploidy female parents have enhanced protein synthesis and chromatin remodeling (Figure 3C). However, later in the manuscript, the authors suggest alternative splicing and protein processing could be responsible for this hexaploid female parent effect in pentaploids. Although it would require functional characterization of some of the candidate genes to decipher the higher ploidy female effect; however, this discrepancy should be addressed in the manuscript text.

The authors suggest that alternative splicing is affected in reciprocal crosses, and it increases with the developmental progression in wheat embryos. They also argue that this could be the reason for perturbed gene expression and effects of female parent ploidy in pentaploids. These are interesting data; however, they need further confirmation and validation. Differences in alternative splicing need to be confirmed for some of the candidate transcripts (maybe 5-10) in 3-4 embryonic stages (if not all) in the reciprocal crosses, particularly involving pentaploid F1s to rule out these not mere statistical artifacts. This for example can be done by RT-PCR followed by amplicon sequencing for splice variants.

Although no consensus on maternal and paternal contributions to gene expression in Arabidopsis zygotes has been reached yet due to the controversy surrounding equi-parental contributions (Nordine & Bartel, Nature 2012; Zhao et al., Dev. Cell 2019) vs. maternal bias (Autran et al., Cell 2011; Leon et al., Nature 2014; Alaniz-Fabián bioRxiv 2020), in cereal crops a general consensus is that gene expression in zygotes is maternally biased (Anderson et al., 2017, Dev Cell; Chen et al., 2017, Plant Cell). Thus, it is not surprising to see more MEGs than PEGs in the E1 stage. The interesting observation from these data is that some of the imprinted genes persisted during E2 and E3 stages. How these imprinted genes affect seed development in interspecific wheat hybrids needs to be investigated in the future. However, I could not make out from the results whether these genes are actually imprinted or just have maternal/paternal bias in their expression? Authors need to make this distinction clear.

One minor comment: figure legends could be more elaborated, some of the figure legends had very limited details and were difficult to interpret.

Overall, the experiments and analyses carried out are of high quality and the gene expression data generated in this study will serve as a great resource and reference for not only interspecific pentaploid F1 hybrids embryos but wheat in general. The data from this study have the potential to enhance our understanding of sterility and incompatibility associated with pentaploid F1 hybrids in wheat and harness their potential for agronomic gains.

Dear Reviewers:

We thank you for the invaluable comments and suggestions of ways to improve our manuscript. We have carefully studied the critiques and revised the manuscript accordingly. A revised version that highlights all revisions in Yellow was also included in the resubmission. Our specific responses and revisions are listed below:

Reviewers' comments:

Reviewer #1 (Remarks to the Author):

The authors investigated genome-wide maternal and paternal contributions to polyploid grain development and analyzed transcriptomes from zygote to mature stage embryos as well as two endosperm stages derived from reciprocal crosses between tetraploid and hexaploid wheats. Results are novel and attractive, especially the homoeologous gene expression of D genome in the pentaploid and the identification of imprinted genes. However, some evidence needs to be provided to support the conclusion, and a comprehensive comparative analysis of the imprinted genes identified by the authors with previous results is required. Details are as follows:

Point 1:

Q1: In the introduction, the authors demonstrated the background and significance of this study. However, the introduction of background is not enough, for instance, Lines 60-61 “negative or additive effects” for details and hybridization barriers and sterility in polyploid wheat for details.

Response: Thank you for the comments. We have provided details on hybridization barriers and sterility in polyploid wheat, as well as transcriptome studies in F1 hybrids, as detailed below (Line 66-102). All relevant references were added in the Reference section.

“Pentaploid interspecific hybrids between wild emmer and common wheat cultivars produced highly sterile offspring (fertility of 1%–2%) (Grana and Gerechter-Amitai, 1974), although such fertility issues can usually be overcome with further crosses between the hybrids and hexaploid wheat cultivars to hasten the recovery of euploid progeny ($2n=42$) with introgressed genes (Lanning et al. 2008; Martin et al., 2011).

Studies on interspecific hybridization of hexaploid and tetraploid wheat have found that the use of higher ploidy species as the maternal parent improves seed set and germination (Lanning et al. 2008; Xie and Nevo, 2008; Martin et al., 2011). Most interspecific hybridizations between different ploidy levels use the hexaploid wheat as the female parent (Wang et al. 2005; Lanning et al. 2008; Martin et al., 2011). Although an interspecific wheat cross using the tetraploid durum wheat as the female parent and the hexaploid bread wheat as the male parent has been developed, none of

the F2 progeny contained a complete set of the seven D genome chromosomes (Wang et al., 2005). Therefore, understanding the barriers to hybridization and cross-directional effects is critical for developing successful interspecific wheat hybrids.

In plants, interspecific hybridization can lead to genomic accommodations such as chromosome reconstitution (Lai et al. 2005), nuclear-cytoplasmic interactions (Michalak et al., 2013), non-additive gene expression, altered alternative splicing (AS) (Yu et al., 2020), and changes in epigenetic regulation (Wang et al., 2014). Changes in gene expression and genome dominance have been observed in hybridization and polyploidization events in plants and animals (Adams 2007; Li et al. 2014; Ren et al., 2019). The reunion of two diverged genomes into a common nucleus during hybridization can result in negative or additive effects on transcript levels in different tissues, growth stages and response to abiotic stresses (Qi et al., 2012; Moran et al., 2021). For example, transcriptome analyses in seedlings of maize revealed alterations in the expression of a number of genes in F1 hybrids compared to their inbred parents (Zhou et al., 2019). Moreover, additive gene expression patterns and biased allele abundance appear to be prevalent in maize hybrids (Zhou et al., 2019). In allopolyploid species such as cotton and wheat, hybridization often leads to imbalances in homeolog gene expression and homeolog silencing, derived from inter- and intra-subgenome interactions (Grover et al., 2012; Yoo et al., 2013; Li et al., 2014). Homeolog expression bias is proposed to be regulated by cis- and trans-acting regulatory interactions among genes and alleles in hybrids and allopolyploids (Hu and Wendel, 2019). However, the effects of merging two regulatory networks into one genetic system and the impact on genome-wide allelic expression changes in polyploidy species remain largely unexplored.”

Q2.1: Identified imprinted genes: In Results, embryo-imprinted genes were defined with “the ratio of maternal or parental reads to the total was greater than 0.7 for embryo stage” were analyzed and identified imprinted genes (Lines 430-431). However, authors identified embryo-imprinted genes in the Experimental procedures by using “ $\geq 85\%$ of total SNP-containing reads that were maternally derived or $\geq 70\%$ that were paternally derived were identified as maternally bias imprinted genes in embryos” (Lines 738-740). Which ratio was used for identifying the embryo-imprinted genes?

Response: *Thank you for this comment. Considering that the embryo receives one “dose” of both the maternal and paternal genome, while the endosperm receives two doses of the maternal genome and one dose of paternal genome, a ratio of 0.7 was used for both maternally and paternally biased gene identification in embryos while a ratio of 0.85 for maternal reads and 0.6 for paternal reads in endosperm tissues. Thus, we have revised the following sentence in Methods Section for clarity (line 781-786): “Paternally derived in two reciprocal crosses of both biological replicates (with a minimum of 10 SNP-associated reads per cross) were identified as imprinted genes. Genes with χ^2 goodness-of-fit test ($FDR < 0.01$) and $\geq 70\%$ of total SNP-containing reads that were maternally or paternally derived were identified as maternally or*

paternally bias imprinted genes in embryos. For endosperm, the ratio is 0.85 for maternal reads and 0.6 for parental reads (FDR<0.01)."

Q2.2 Confirm the asymmetric of imprinted genes: In this study, a lot of imprinted genes were identified during embryogenesis in polyploid wheats. However, it is unclear that these imprinted genes showed allelic imbalance and asymmetric expression in fact. Please randomly select some candidate imprinted genes to confirm the asymmetric expression by using the RT-PCR sequencing or allele specific RT-PCR which is done for example in Chen et al., 2018.

Response: As suggested, we have selected several candidate imprinted genes and confirmed the asymmetric expression by using the ddPCR analysis. These data have been added to the revised version of the manuscript as shown in a new Figure 7.

A paragraph about the validation of imprinting genes was included in the revised manuscript (line 494-505):

"Selected imprinted genes were also confirmed in embryo and endosperm. Allele-specific fluorescent probes were designed based on the SNPs identified between two parents of SxA and AxS, respectively. Droplets from 5'-labeled with 6-carboxyfluorescein (FAM) and 6-carboxy-2, 4, 4, 5, 7, 7 hexachlorofluorescein succinimidyl ester (HEX) probes were clustered in the 2-D droplet amplitude plots and the ratio of FAM and HEX droplets were calculated to compare the ratios of maternal or parental reads from RNAseq (Figure 7C and 7D). The results showed that the ratios of maternal reads to total reads from RNAseq were consistent with those from ddPCR droplets to the total droplets in tested genes at embryos in E3 and endosperm in E8 (Figure 7E). Thus, the expression pattern of selected MEGs and PEGs was independently validated. Together, our results indicated an overall reliability of the differential expression analysis, AS events and imprinting gene identification."

In addition, selected DEGs and AS events were also confirmed by ddPCR.

Q2.3 Discussion the imprinted genes in polyploid wheat: The discussion of imprinted genes in this study is not depth. Yang et al (2018) suggested that genomic imprinting was evolutionarily conserved during wheat polyploidization by identified imprinted genes in diploid, tetraploid, and hexaploid wheat, respectively. Molecular evolution and preferential expression of imprinted genes can be discussed in this study by compared with the imprinted genes reported by Yang et al (2018), and these comparisons are favorable to improve the credibility of the gene imprinting status in this study.

Response: Most previous studies identified imprinting genes from endosperm only, including Yang et al's (2018) study. As suggested by the reviewer, we have compared the imprinting genes identified in our study with that of Yang et al., 2018. Yang et al (2018) identified a total of 372 imprinted genes using TGACv1 as the reference genome. After converting the TGACv1 IDs to IWGSC1.1 IDs using the ID History

Converter tool (http://plants.ensembl.org/Triticum_aestivum/Tools/IDMapper/). After removing duplicated IDs , a total of 285 genes with corresponding IWGSC1.1 ids were retained. Of these 285 genes, 62 genes overlapped with the imprinted genes identified in endosperm in this study as shown in Data S15. Based on the above analysis, we added the following sentence to the Discussion (Line 631-637):

“A previous study identified a total of 372 imprinted genes in developing endosperm of diploid, tetraploid and hexaploid wheat (Yang et al., 2018), while we detected 658 from embryos and endosperm of all four cross pairs in this study. More imprinted genes were identified from embryos in this study and there were 62 genes commonly identified in endosperm (Data S15). The discrepancy may be due to different tissue types and development stages collected in this study.”

Q3: About the plant material ‘strongfield’. Lines 97/1023 showed “Strong Field”, Lines 607/721 showed “strongfield”. Please correct it.

Corrected

Q4: Minor points:

Line 67—please correct “Triticum turgidum ssp. Durum” into “Triticum turgidum ssp. durum”.

Corrected

Lines 84/248/508/667/703—abbreviation AS should in here.

Corrected

Line 104—“two stages of endosperm (transition and leaf late stage) ”, while in DataS1, two stages of endosperm are early and late.

Corrected

Lines 521-523—“Comparisons between reciprocal crosses and their parents showed similar numbers of AS events between pentaploids and hexploids while tetraploids showed AS events”. This sentence sounds somewhat contradictory.

This sentence has been removed.

Line 578—“maternally- and paternally imprinted genes”, remove “-”.

Corrected

Lines 626/725—Please, correct “Dobin et al., 2012” into “Dobin et al., 2013”.

Corrected

Lines 635/638—Please, correct “Trapnell et al., 2012” into “Trapnell et al., 2010”.

Corrected

Line 669—please, correct “were obtained from Xiang et al., 2019” into “were obtained from Xiang et al (2019)”.

Corrected

Lines 678/708—abbreviation TPM should in here.

Corrected

Q5: Lines 719-722—“raw reads of the four parents collected at the sample developmental stages (A, AC Barrie; C, Commander; S, strongfield; Z, Chinese Spring) (Xiang et al., 2019) were used for SNP and allele-specific gene expression analysis”. However, ‘Commander’ and ‘Chinese Spring’ were not found in Xiang et al., 2019.

Response: In our previous studies, we generated RNAseq data from AC Barrie and Strong Field (Xiang et al., 2019), while RNASeq data from Commander and Chinese Spring were from Gao et al (2021). The RNAseq data was collected at the sample developmental stages as that of reciprocal crosses. Therefore, these data from parental lines were used in this study for SNP and allele-specific genes expression analysis.

The manuscript has been revised accordingly (Line 760-763):

“Raw RNA sequencing data of reciprocal crosses generated in this study and raw reads of the four parents collected at the sample developmental stages (A, AC Barrie; C, Commander; S, Strong Field; Z, Chinese Spring) (Xiang et al., 2019; Gao et al., 2021) were used for SNP and allele-specific gene expression analysis. “

Q6:

References 13/60. Authors’ name should be corrected.

Corrected

References 34/55 are mentioned in the References but are missing in the text.

References 34 was cited in the main text in Line 337-338 as below:

“Consistent with previous studies (Ma et al., 2014), rearrangements were found on chromosome 4A, 5A and 7B (Figure S21).”

References 55 was cited in the main text in Line 706-707 as below:

“Genes involved in AS pathways were compiled from SRGD datase (<http://www.plantgdb.org/SRGD/>) (Wang and Brendel, 2004).”

Both references were included in the Reference section.

Reviewer #2 (Remarks to the Author):

This study analyzed the differentially expressed genes, alternative splicing and parental allele expression patterns of embryos and endosperms in the reciprocal crosses of hexaploidy wheat, tetraploid wheat as well as pentaploid wheat, and the authors also compared homeologous gene expression patterns in different tissues of diverse samples. The study provides a large set of data for developing embryo and endosperms in wheat. However, the data presentation needs lots of improvement to fully support the conclusion. The following suggestions may help to improve the manuscript, which is encouraged to resubmit with revisions.

Q1: The title is not suitable for the manuscript.

Response: We thank the reviewer for the careful consideration of our manuscript and the constructive feedback. We have changed the title to “Asymmetric gene expression in grain development of reciprocal crosses between tetraploid and hexaploid wheats”

Q2: The authors should provide phenotypic analysis of developing endosperm and embryo between interploidy reciprocal crosses, and it is better to analyze the association between gene expression and phenotypic variation.

Response: Thank you for the comment. We have examined embryo and endosperm development in reciprocal crosses of hexaploid and tetraploid wheats and added a new figure (Figure S1) in the Supplemental files. The developing embryos of Ac Barrie x Strong Field (AxS) and Strong Field x Ac Barrie (SxA) were used as representatives. Micrographs depicting the developmental progression of the wheat embryo from two cell zygote to maturity were obtained. As shown in Figure S1, we did not observe clear morphological differences in the developing embryo and endosperm between reciprocal cross AxS and SxA were subtle over the seven stages of embryo development. Endosperm at the transition stage (E8) and leaf late stage (E9) was also examined in both parents and reciprocal F1 hybrids in AC Barrie, Strongfield, AxS and SxA. It was not feasible to identify the phenotypic variation between reciprocal crosses.

Q3: The authors included too much information in the manuscript, e.g. DEGs, AS genes, homeologous gene variation and parental allele expression (imprinting). It might be better to focus on two aspects, and provide more detailed information.

Response: Thanks for your suggestion. Changes in DEGs, AS and imprinting genes between or within reciprocal cross pairs all relate to transcriptional regulation. Therefore, we focused on these aspects. We considered this study to provide a comprehensive analysis that presents the transcriptional changes during embryogenesis between interspecific crosses.

Q4: The validation of imprinting, AS, DEG is needed, especially, previous data indicated that imprinted genes is mainly occurred in endosperm and very few were identified in embryo. The author identified a set of imprinted genes in wheat embryo,

should provide confident evidence for this finding.

Response: Thank you for the feedback. We have now validated a few DEGs, AS as well as imprinting genes using ddPCR. For the genes tested, the RNAseq results are consistent with ddPCR data, supporting the reliability of the RNAseq data. A paragraph and a new figure (Figure 7) about the validation was added to the manuscript as described below (line 476-505):

“Validation of DEGs, AS events and imprinting genes using digital PCR (ddPCR)

To validate our results from DEG, AS and imprinting gene analyses, we selected several genes and transcripts for droplet digital PCR (ddPCR) assays using specifically designed primers and probes. For the validation of DEGs, gene-specific primers were designed for the target genes to assess gene expression changes in different stages. The expression levels of three DEGs, XLOC_061322, XLOC_123260 and XLOC_114665 at stage E4, E5 and E6 from the cross between Strong Field and Ac Barrie (SxA) were quantified and compared (Figure 7A). Both RNAseq and ddPCR showed similar log₂FoldChange values between E5 and E4 as well as E6 and E4, suggesting good correlation between the two techniques (Pearson correlation value = 0.976).

For the validation of AS events, primers targeting one or multiple different transcripts were designed and the expression of each transcript was calculated. AS site-spanning primers were designed with two fluorescent probes and 2–3 base pair mismatches to ensure accurate detection and quantification of transcript isoforms from each homoeologous gene (see details in Methods). We examined the expression level of four AS events at stage E5 and E6 in the reciprocal cross pair of SxA and AxS using ddPCR. Our results showed a high correlation between RNAseq and ddPCR (Pearson correlation value = 0.839).

Selected imprinted genes were also confirmed in embryo and endosperm. Allele-specific fluorescent probes were designed based on the SNPs identified between two parents of SxA and AxS, respectively. Droplets from 5'-labeled with 6-carboxyfluorescein (FAM) and 6-carboxy-2, 4, 4, 5, 7, 7 hexachlorofluoresce in succinimidyl ester (HEX) probes were clustered in the 2-D droplet amplitude plots and the ratio of FAM and HEX droplets were calculated to compare the ratios of maternal or parental reads from RNAseq (Figure 7C and 7D). The results showed that the ratios of maternal reads to total reads from RNAseq were consistent with those from ddPCR droplets to the total droplets in tested genes at embryos in E3 and endosperm in E8 (Figure 7E). Thus, the expression pattern of selected MEGs and PEGs was independently validated. Together, our results indicated an overall reliability of the differential expression analysis, AS events and imprinting gene identification. ”

Imprinting genes have been previously identified in embryos in maize, Arabidopsis and rice (Jahnke and Scholten, 2009; Nodine and Bartel, 2012; Raissig et al., 2013; Meng et al., 2018). Our study identified a number of imprinting genes in wheat embryos. Selected MEGs and PEGs in embryos at E3 stage as well as in endosperm at E8 stage were further validated using ddPCR, and support the reliability of the

imprinting genes identified through RNASeq in this study.

Q5: The authors should provide more detailed information in the methods section. 1) how do the authors analyze the reads in different biological replicates when identifying imprinted genes, the criteria of >5 SNP-associated reads is not sufficient; 2) when analyzing gene expression patterns of endosperm in pentaploid wheat, how do the author normalize the sequencing reads, since the reciprocal crosses have different sets of chromosomes; 3) The author declared two replicates of samples were collected, are they separately sequenced or mixed for sequencing? Should be stated clearly.

Response: *Thanks for raising these points. As suggested, we have re-analyzed the imprinting genes as below:*

1. SNP identification and imprinting gene analysis was performed according to the protocol from Picard and Gehrung (2020).

According to the reviewer's suggestion, we have re-analyzed each of the two biological replicates separately from 216 BAM files for SNP identification. A minimum of 10 SNP-associated reads in each reciprocal cross was used for the identification of imprinted genes. Genes identified as imprinting in both replicates (minimum reads > 10 and FDR < 0.01) were categorized as MEGs or PEGs. With the new analysis, fewer imprinting genes were identified. We have changed all relevant descriptions in the main text (highlighted in yellow), Materials and Methods and re-generated all related data and figures, including Figure 6, Data S14, Data S15, Figure S26, Figure S27 and Figure S28 .

2. For endosperm, it was not possible to compare the normalized counts between different sets or size of genomes. Sample from two stages were only used for the identification of imprinting genes since the endosperm tissue has distinct chromosome sets. For the imprinting genes, SNP associated reads from parents were obtained and analyzed (Figure 6). For embryos, the pentaploids have the same number of chromosome sets (AABBD). For endosperm tissues, pentaploids have AAABBBDD when the female parent is hexaploid and AAABBBDD when the female parent is tetraploid. DEGs, AS and homeologous expression analysis were performed with embryo tissues as shown in Figure 1 to Figure 5. Sequencing reads were normalized as TPM values for comparisons.

3. The two replicates used in this study were two biological replicates which were collected and sequenced separately.

Q6. The writings needs improvement.

Response: *The manuscript has been revised by a native English speaker and scientist.*

Reviewer #3 (Remarks to the Author):

Interspecific pentaploid F1 hybrids generated by crossing hexaploid and tetraploid wheat species have distinctive genetic variability due to chromosomal reconstitution which has great potential to improve several agronomic traits like disease resistance, abiotic stress tolerance, and grain quality. However, these interspecific pentaploid F1s suffer from sterility, reduced seed set, and pollen incompatibility, besides other deficiencies. In this manuscript by Jia et al., authors investigated gene expression in pentaploid F1s derived from crosses between two hexaploid wheat species - Chinese Spring and AC Barrie, and two tetraploid varieties - Strong Field and Commander. The authors analyzed seven stages from zygote to mature embryo during embryonic development and two endosperm stages. Reciprocal crosses were carried out to look into parental contributions to gene expression during these stages of seed development and observe the parent-of-origin-dependent gene expression as well.

First authors established expression profiles in different ploidy levels for different genome combinations (hexaploidy = AABBDD, pentaploidy = AABBDD, and tetraploidy = AABB) as a reference for investigating transcriptomes in their reciprocal crosses and developmental stages. As one can expect, gene expression in hexaploids with ABD genomes was closer to pentaploids (with the same ABD genomes) than tetraploids with just AB genomes. Comparisons were made between reciprocal crosses and different samples. Analysis of DEGs showed an increase with temporal development; hexaploids and pentaploids have more DEGs compared to tetraploids, and almost 60% of DEGs were shared between each reciprocal cross pair.

Q1: Analysis from other studies has led to an interesting observation that using the higher ploidy species as the female parent can lead to reduced sterility in the wheat F1 pentaploids as the authors have correctly mentioned in the introduction. The authors show that the pentaploids with higher ploidy female parents have enhanced protein synthesis and chromatin remodeling (Figure 3C). However, later in the manuscript, the authors suggest alternative splicing and protein processing could be responsible for this hexaploid female parent effect in pentaploids. Although it would require functional characterization of some of the candidate genes to decipher the higher ploidy female effect; however, this discrepancy should be addressed in the manuscript text.

Response: Thank you for pointing out the discrepancy. We identified the differentially expressed genes (DEGs) in F1 hybrids between each cross pair. After summing the number of DEGs, it was clear that pentaploids yields almost double the number of DEGs as the hexaploid or tetraploid crosses. The DEGs were further categorized into different biological pathways. A number of up-regulated genes were involved in protein synthesis and chromatin remodeling pathways in both AxCvsCxA and AxSvsSxA, suggesting a higher level of protein synthesis and chromatin remodeling in pentaploids whose female parent is hexaploid.

The combination of divergent genomes within a hybrid might induce a “stress” signal that affects the transcription of many genes exclusively in the hybrid. We therefore

further investigated genes that are consistently highly or lowly expressed in F1 hybrid across all embryo stages via paired ANOVA tests for each DEG. 3,124 genes were commonly identified in the pentaploids, while in AxZ vs ZxA and CxS vs SxC where parents have the same ploidy levels, only 62 and 14 genes were identified (FDR < 0.05). The results suggested a distinct gene expression pattern exclusively in the F1 hybrid between reciprocal crosses with different ploidy levels. This set of genes might be specifically related to cross-direction effects. GO enrichment analysis also support the involvement of mRNA splicing, protein transport, etc. Therefore, both transcriptional (chromatin remodeling and mRNA splicing) and post-transcriptional processes (protein synthesis and protein transport) were hybridization effects.

The results and conclusions were from different angles of analysis and we have since summarized the results as below (Line 262-268):

“Together, our results demonstrate extensive changes in transcriptional regulations in F1 hybrid embryos after hybridization, and to a greater extent in crosses between wheats with different ploidy levels. Moreover, analysis of cross-directional effects suggests alternative splicing, protein processing and chromatin remodeling are important processes in embryogenesis in pentaploids with a hexaploid female parent, while protein modification and transport are involved in regulating embryogenesis in pentaploids with a tetraploid female parent. “

Q2: The authors suggest that alternative splicing is affected in reciprocal crosses, and it increases with the developmental progression in wheat embryos. They also argue that this could be the reason for perturbed gene expression and effects of female parent ploidy in pentaploids. These are interesting data; however, they need further confirmation and validation. Differences in alternative splicing need to be confirmed for some of the candidate transcripts (maybe 5-10) in 3-4 embryonic stages (if not all) in the reciprocal crosses, particularly involving pentaploid F1s to rule out these not mere statistical artifacts. This for example can be done by RT-PCR followed by amplicon sequencing for splice variants.

Response: We have validated a few DEGs, AS, as well as imprinting genes using ddPCR. Our results showed that the RNAseq results were consistent with those from ddPCR in tested genes, thus showed an overall reliability of RNAseq data. A paragraph and a new figure(Figure 7) about the validation was added to the manuscript as described as below (line 476-505):

“Validation of DEGs, AS events and imprinting genes using digital PCR (ddPCR)

To validate our results from DEG, AS and imprinting gene analyses, we selected several genes and transcripts for droplet digital PCR (ddPCR) assays using specifically designed primers and probes. For the validation of DEGs, gene-specific primers were designed for the target genes to assess gene expression changes in different stages. The expression levels of three DEGs, XLOC_061322, XLOC_123260 and XLOC_114665 at stage E4, E5 and E6 from the cross between Strong Field and Ac Barrie (SxA) were quantified and compared (Figure 7A). Both RNAseq and ddPCR showed similar log₂FoldChange values between E5 and E4 as well as E6 and E4, suggesting good correlation between the two techniques (Pearson correlation value =

0.976).

For the validation of AS events, primers targeting one or multiple different transcripts were designed and the expression of each transcript was calculated. AS site-spanning primers were designed with two fluorescent probes and 2–3 base pair mismatches to ensure accurate detection and quantification of transcript isoforms from each homoeologous gene (see details in Methods). We examined the expression level of four AS events at stage E5 and E6 in the reciprocal cross pair of SxA and AxS using ddPCR. Our results showed a high correlation between RNAseq and ddPCR (Pearson correlation value = 0.839).

Selected imprinted genes were also confirmed in embryo and endosperm. Allele-specific fluorescent probes were designed based on the SNPs identified between two parents of SxA and AxS, respectively. Droplets from 5'-labeled with 6-carboxyfluorescein (FAM) and 6-carboxy-2, 4, 4, 5, 7, 7 hexachlorofluorescein succinimidyl ester (HEX) probes were clustered in the 2-D droplet amplitude plots and the ratio of FAM and HEX droplets were calculated to compare the ratios of maternal or parental reads from RNASeq (Figure 7C and 7D). The results showed that the ratios of maternal reads to total reads from RNASeq were consistent with those from ddPCR droplets to the total droplets in tested genes at embryos in E3 and endosperm in E8 (Figure 7E). Thus, the expression pattern of selected MEGs and PEGs was independently validated. Together, our results indicated an overall reliability of the differential expression analysis, AS events and imprinting gene identification. ”

Q3: Although no consensus on maternal and paternal contributions to gene expression in Arabidopsis zygotes has been reached yet due to the controversy surrounding equi-parental contributions (Nodine & Bartel, Nature 2012; Zhao et al., Dev. Cell 2019) vs. maternal bias (Autran et al., Cell 2011; Leon et al., Nature 2014; Alaniz-Fabián bioRxiv 2020), in cereal crops a general consensus is that gene expression in zygotes is maternally biased (Anderson et al., 2017, Dev Cell; Chen et al., 2017, Plant Cell). Thus, it is not surprising to see more MEGs than PEGs in the E1 stage. The interesting observation from these data is that some of the imprinted genes persisted during E2 and E3 stages. How these imprinted genes affect seed development in interspecific wheat hybrids needs to be investigated in the future. However, I could not make out from the results whether these genes are actually imprinted or just have maternal/paternal bias in their expression? Authors need to make this distinction clear.

Response: *Thanks for the comment. Imprinting gene identification was performed according to the protocol from Picard and Gehring (2020). Although both imprinted genes and parental biased genes were identified, only imprinted genes (MEGs and PEGs) were presented and analyzed in the manuscript (Data S15).*

In addition, we have re-analyzed each of the two biological replicates separately from 216 BAM files for SNP identification according to one of the reviewer's suggestion. A minimum of 10 SNP-associated reads in each reciprocal cross was used for the identification of imprinted genes. Genes identified as imprinting in both replicates

(minimum reads > 10 and FDR < 0.01) were assigned as MEGs or PEGs. With the new analysis, fewer imprinting genes were identified. We have since changed all relevant descriptions in the main text (highlighted in yellow) and re-generated all related data and figures, including Figure 6, Data S14, Data S15, Figure S26, Figure S27 and Figure S28.

As described in the Methods section, imprinted genes were identified based on gene-associated SNPs and read counts, as detailed below (Line 778-786):

“Allele-specific SNPs between the two parents were identified and subsequently used for SNPs identification in F1 hybrids in vcf files. The paternal SNPs located in the gene were processed into allele-specific, per-gene counts using the identified SNPs between male and female parents, calculated by custom Python scripts. Paternally derived in two reciprocal crosses of both biological replicates (with a minimum of 10 SNP-associated reads per cross) were identified as imprinted genes. Genes with χ^2 goodness-of-fit test (FDR < 0.01) and $\geq 70\%$ of total SNP-containing reads that were maternally or paternally derived were identified as maternally or paternally bias imprinted genes in embryos. For endosperm, the ratio is 0.85 for maternal reads and 0.6 for parental reads (FDR<0.01).”

In addition, selected MEGs and PEGs genes were validated respectively using ddPCR (Figure 7E).

Q5: One minor comment: figure legends could be more elaborated, some of the figure legends had very limited details and were difficult to interpret.

Response: *Thanks for the feedback. We have rewritten the figure legends, and added more information to the Methods sections as highlighted in yellow in the main text.*

Overall, the experiments and analyses carried out are of high quality and the gene expression data generated in this study will serve as a great resource and reference for not only interspecific pentaploid F1 hybrids embryos but wheat in general. The data from this study have the potential to enhance our understanding of sterility and incompatibility associated with pentaploid F1 hybrids in wheat and harness their potential for agronomic gains.

Response: *Thanks for the positive comments and feedback.*

Reviewers' comments:

Reviewer #1 (Remarks to the Author):

The reviewer's questions were well resolved, and the quality of the MS has also been improved. I think that the MS may be accepted for publication in CB.

Reviewer #2 (Remarks to the Author):

The authors have addressed most of my concerns, however, I still have several questions about the manuscript

Q2: The authors should provide phenotypic analysis of developing endosperm and embryo between interploidy reciprocal crosses, and it is better to analyze the association between gene expression and phenotypic variation.

I am not convinced by the phenotyping analysis of interploidy crosses. It is reported that more than 99% of kernels were highly defective in a maternal-excess cross ($4n \times 2n$), and ~9.7% plump kernels with normal phenotypes were observed in a paternal-excess cross in maize ($2n \times 4n$; Leblanc et al., 2002). Similarly in rice, an interploidy cross with paternal excess strongly affects seed starch accumulation and gives rise to progeny with non-germination percentage; and both paternal- and maternal-excess interploidy crosses generate smaller kernels (Sekine, 2013; Zhang et al., 2016). Whereas in Arabidopsis, a paternal-excess cross generates heavier seeds compared to a maternal-excess cross and a balanced cross (Scott et al., 1998).

The provided pictures cannot support the conclusion that no differences of endosperm were observed between the two reciprocal crosses, should provide pictures of whole seed and endosperm, and perform statistical analysis.

Q3: The authors included too much information in the manuscript, e.g. DEGs, AS genes, homeologous gene variation and parental allele expression (imprinting). It might be better to focus on two aspects, and provide more detailed information

As the authors prefer to retain all the transcriptional analysis, should be very careful with the calculation methods. For example, the chromosome constitution of subgenome D are different between hexaploidy and tetraploid interploidy crosses, it does not make sense to identify differentially expressed genes in subgenome D between each other, since the author also admit that it is difficult to normalize the data. Similar situation occurs to identification of differentially expressed genes involved in AS.

Moreover, when identifying imprinted genes, it would be better to remove sequencing reads mapping to subgenome D first and only consider reads for subgenome A and B.

In line 460-462, it is stated that "Due to the univalent D genome in pentaploids and the lack of a D genome in tetraploids, MEGs and PEGs were identified in the A and B subgenomes in pentaploids and tetraploid", however, in line 470-472, it is described that "In contrast, imprinting triads in pentaploids tended to be expressed in A and B subgenomes and many genes were categorized as D suppressed (Figure 6F and Figure S2". The descriptions are contradicted with each other. And comparison of D subgenome related genes are not suitable as mentioned above.

Q4: The validation of imprinting, AS, DEG is needed, especially, previous data indicated that imprinted genes is mainly occurred in endosperm and very few were identified in embryo. The author identified a set of imprinted genes in wheat embryo, should provide confident evidence for

The authors should specify the chromosomal location of the selected genes, as mentioned above, D subgenome related genes should not be considered during the analysis.

Reviewer #3 (Remarks to the Author):

The authors have answered all the questions and addressed the concerns I had during the initial review, and they have modified the manuscript accordingly. I have no further comments. I congratulate them on their achievement to improve the report. I am confident the manuscript has merit for publication in the 'Communications Biology' and should now be accepted for publication.

Manuscript ID: COMMSBIO-22-0433B

Dear Reviewers:

We thank you for the valuable comments and constructive suggestions to improve our manuscript. We have carefully considered the reviews and revised the manuscript accordingly. A revised version with all revisions highlighted in yellow is included in our resubmission. Our point by point responses and revisions are detailed below.

Referee expertise:

Referee #1: Plant functional genomics

Referee #2: Polyploidy and plant epigenetics

Referee #3: Genetics of seed development

Reviewers' comments:

Reviewer #1 (Remarks to the Author):

The reviewer's questions were well resolved, and the quality of the MS has also been improved. I think that the MS may be accepted for publication in CB.

Response: We thank this reviewer for critical assessment and approval of our revised manuscript. The reviewer's constructive feedback improved the quality of the manuscript.

Reviewer #2 (Remarks to the Author):

The authors have addressed most of my concerns, however, I still have several questions about the manuscript

Q2: The authors should provide phenotypic analysis of developing endosperm and embryo between interploidy reciprocal crosses, and it is better to analyze the association between gene expression and phenotypic variation.

I am not convinced by the phenotyping analysis of interploidy crosses. It is reported that more than 99% of kernels were highly defective in a maternal-excess cross ($4n \times 2n$), and $\sim 9.7\%$ plump kernels with normal phenotypes were observed in a paternal-excess cross in maize ($2n \times 4n$; Leblanc et al., 2002). Similarly in rice, an interploidy cross with paternal excess strongly affects seed starch accumulation and gives rise to progeny with non-germination percentage; and both paternal- and maternal-excess interploidy crosses generate smaller kernels (Sekine, 2013; Zhang et al., 2016). Whereas in Arabidopsis, a paternal-excess cross generates heavier seeds compared to a maternal-excess cross and a balanced cross (Scott et al., 1998).

The provided pictures cannot support the conclusion that no differences of endosperm were observed between the two reciprocal crosses, should provide pictures of whole seed and endosperm, and perform statistical analysis.

Response: We thank this reviewer for additional feedback. As suggested, in the revised version, we have performed additional phenotypic analysis of whole seeds and endosperm between different interploidy reciprocal crosses (Figure S1B). As shown in Figure S1B, the seed weight, seed length, seed width, seed perimeter and seed area are significantly affected (reduced) in the mature seeds of tetraploid x hexaploid and

hexaploid x tetraploid crosses, compared to crosses of the same ploidy (tetraploid x tetraploid and hexaploid x hexaploid). In particular, fewer and some abnormal endosperm cells were observed in tetraploid x hexaploid and hexaploid x tetraploid crosses, when compared to the endosperm resulting from tetraploid x tetraploid and hexaploid x hexaploid crosses. These data indicated endosperm cell division, endosperm development and seed starch accumulation are impacted after interspecific hybridization. In our study, the grain size of two tetraploid lines (S and C) was bigger than that of the two hexaploid lines (A and Z); the hybrid grain size was bigger in tetraploid x hexaploid cross than the hexaploid x tetraploid cross, as shown in Figure S1B. Additionally, we noticed a slight difference in the success rate of crosses between interploidy parents. In the controlled growth chamber conditions (16 hours light, 22 °C and 8 hours dark, 18 °C), the emasculating time significantly affected the cross success rate. If the pollination was performed immediately after emasculating, the cross success rate was around 40-50 % for hexaploid x tetraploid and 20-30% for tetraploid x hexaploid, whereas, the rate was 90-95% for hexaploid x tetraploid cross and 85-90% for tetraploid x hexaploid cross when pollination was performed 3 days after emasculating. These results suggest that the A, B and D subgenomes in polyploid wheats provide additional genetic buffers relative to interploidy crosses in Arabidopsis, rice and maize.

Description about the phenotype of hybrid grains can be found in the main text as below (line 122-135):

“The developing embryos and endosperm resulting from each reciprocal cross did not exhibit notable morphology differences at collected stages (Figure S1A). However, statistical analysis of mature F1 seeds of different crosses showed the seed weight, seed length, seed width, seed perimeter and seed area were significantly reduced in tetraploid x hexaploid and hexaploid x tetraploid interspecific crosses, compared to the same ploidy crosses (Figure S1B and S1C). In particular, fewer and some abnormal endosperm cells were observed in tetraploid x hexaploid and hexaploid x tetraploid interspecific crosses, when compared to the endosperm resulting from the same ploidy crosses (Figure S1B), indicating endosperm cell division, endosperm development and seed starch accumulation are impacted after interspecific hybridization. Furthermore, the grain weight always showed higher values both in the interspecific crosses and the same ploidy crosses if the maternal parent has larger seed size than paternal parent in mature seeds (Figure S1B), suggesting an maternal impact on grain development after hybridization. ”

As described above and Figure S1, the visible differences of developing embryos and endosperm between reciprocal crosses were minor and difficult to quantify. In mature F1 seeds, we do observed phenotypic variations after interspecific hybridization. Unfortunately, it is not feasible to correlate the phenotype of mature seeds with gene expression in this study. A deep analysis requires changes at molecular level such as carbohydrates, protein and hormone profiling during grain development. But it is beyond the scope of this study and therefore it is reasonable left for future studies.

Q3: The authors included too much information in the manuscript, e.g. DEGs, AS genes, homeologous gene variation and parental allele expression (imprinting). It might be better to focus on two aspects, and provide more detailed information

As the authors prefer to retain all the transcriptional analysis, should be very careful with the calculation methods. For example, the chromosome constitution of subgenome D are different between hexaploidy and tetraploid interploidy crosses, it does not make sense to identify differentially expressed genes in subgenome D between each other, since the author also admit that it is difficult to normalize the data. Similar situation occurs to identification of differentially expressed genes involved in AS.

Moreover, when identifying imprinted genes, it would be better to remove sequencing reads mapping to subgenome D first and only consider reads for subgenome A and B. In line 460-462, it is stated that “Due to the univalent D genome in pentaploids and the lack of a D genome in tetraploids, MEGs and PEGs were identified in the A and B subgenomes in pentaploids and tetraploid”, however, in line 470-472, it is described that “In contrast, imprinting triads in pentaploids tended to be expressed in A and B subgenomes and many genes were categorized as D suppressed (Figure 6F and Figure S2)”. The descriptions are contradicted with each other. And comparison of D subgenome related genes are not suitable as mentioned above.

Responses: Thank you for the points. We are aware of this problem from the beginning of our analysis and to avoid identifying DEGs and DAS between samples with different ploidy levels.

In embryos, the chromosome numbers are identical in F1 progeny between reciprocal pairs. In endosperm, the chromosome numbers are different in F1 progeny after hybridization between different polyploidy levels. For example, hybridization between hexaploids produces an AAABBBDDD endosperm genome and AABB between tetraploids. Hexaploid and tetraploid interploidy crosses generate AAABBBDD pentaploids when the maternal parent is a hexaploid and AAABBBDD when the maternal parent is tetraploid.

As illustrated in the above figure, F1 embryos contain the same number of chromosomes between reciprocal crosses, while the endosperm generated has

different set of chromosomes. Therefore, to avoid problems during normalization and quantification of gene expression between hexaploidy and tetraploid interploidy crosses as well as cross-sample comparisons, we focused on the identification of DEGs (as stated in line 182-185, line 230-234 and line 255-258) and AS (as stated in line 288-290) **only in embryos (E1 to E7)** as demonstrated in Figure 2, Figure 3 and Figure 4. A detailed explanation can be found below.

(1) For DEGs, comparisons were only performed between samples with the same number of chromosomes.

(a) Stage-DEGs were identified in embryos through comparisons between stages within each of the eight crosses. Description can be found in line 182-185 as below:

“We first evaluated differential expression in each cross during embryo development. Genes with more than one TPM in at least one of all samples were considered expressed. Differentially expressed genes (DEGs) in each embryo sample were identified through comparisons with two-cell stage (E1). ”

(b) Sample-DEGs were identified in embryos between reciprocal pairs at each stage. Description can be found in line 230-234 and line 255-258 as below:

line 230-234 : “A total of 45,774, 38,809, 21,387 and 25,313 DEGs were identified in embryos from AxC vs CxA; AxS vs SxA; CxS vs SxC; and AxZ vs ZxA crosses, respectively (Figure 3A and Data S6). Across seven embryo stages, more DEGs were identified in reciprocal crosses between tetraploid and hexaploid species (AxC vs CxA and AxS vs SxA) than in crosses between species with the same ploidy levels (Figure 3B).”

Line 255-256: “We performed paired ANOVA tests for each DEG to identify genes consistently expressed at higher or lower levels across all embryo stages between each reciprocal pair.”

(c) In addition, the stage-DEGs and sample-DEGs from embryos is described in the Methods section as below (line 718-722):

“Differential expression analysis was performed using DESeq2 to identify stage-DEGs and sample-DEGs (Love et al., 2014). For stage-DEGs, comparisons between each time point against the two-cell stage at embryo stages for each sample were calculated. For sample-DEGs, read counts from each pair of reciprocal crosses were normalized, and comparisons were calculated between F1 embryos at each stage.”

(2) For AS, splicing events were identified in this study as detailed in Figure 3, Data S8, Data S9 and stated in line 288-290 and line 315-317. Differential AS (DAS) was performed between reciprocal crosses **only in embryos (E1 to E7)** at the same stages (Figure 3F, Data S9). The statement can be found as below:

Line 288-290 for AS in embryos: “We quantified PSI, the ratio of a transcript element over the total normalized reads for an AS event, using multiple stages of embryo development.”

Line 315-317 for DAS in embryos: “To explore the AS patterns caused by interspecific hybridization and differential AS (DAS), the magnitude of splicing change (Δ PSI) and

its significance across multiple embryo stages was determined using SUPPA analysis”.

(3) Homeologous triads were classified into seven categories based on their relative proportion of expression levels within each triad in each sample. No direct comparison was made between samples from different ploidy levels.

(4) Regarding the contradictory descriptions, the former sentence is about imprinting genes, while the latter is about homeologous genes. The detailed explanation can be found as below:

The former sentence: “Due to the univalent D genome in pentaploids and the lack of a D genome in tetraploids, MEGs and PEGs were identified in the A and B subgenomes in pentaploids and tetraploid”.

It describes the identification of imprinting genes in A and B genomes in pentaploids and tetraploid. We have now added four columns in Data S15 including chromosome information, start, end, and IWGSC RefSeq 1.1 id information for all imprinting genes identified in this study.

The latter sentence: “In contrast, imprinting triads in pentaploids tended to be expressed in A and B subgenomes and many genes were categorized as D suppressed (Figure 6F and Figure S29) ”.

This sentence describes homeologous triad expression. If one of the three genes in a triad was identified as imprinted, the triad was selected and plotted in the ternary plot. As a result, the ternary plot included imprinted genes and their homeologous genes that were not imprinted from D subgenome. This is why some genes were categorized as D suppressed in the ternary plot, but they were actually not imprinting genes but homeologous genes of imprinting genes.”

To avoid this confusion, we have excluded the ternary plots for imprinting genes and relevant description in the main text and Figure S29.

In summary, the DEGs and DAS were identified in embryos only as mentioned in the main text as well as in figure legends. Imprinted genes were identified from A and B subgenomes in pentaploids and tetraploids.

Q4: The validation of imprinting, AS, DEG is needed, especially, previous data indicated that imprinted genes is mainly occurred in endosperm and very few were identified in embryo. The author identified a set of imprinted genes in wheat embryo, should provide confident evidence for

The authors should specify the chromosomal location of the selected genes, as mentioned above, D subgenome related genes should not be considered during the analysis.

Responses: Thank you for this feedback. In our last revision, we validated few selected DEGs, AS and imprinting genes using ddPCR (Figure 7). For the imprinting

genes, four genes, two from embryos and two from endosperm, were selected for validation. No imprinting genes from D chromosome were selected and the chromosomal information of selected genes are shown as below:

*XLOC_045394;chr2B:571345533-571348162, corresponds to TraesCS2B02G402800 in reference genome. The annotation can be found in Data S3 and updated Data S15. XLOC_045394 was identified as a **MEG from embryos** at E3 stage between AxS and SxA.*

*XLOC_193638;chr7A:154062737-154064285, corresponds to TraesCS7A02G194700 in reference genome. The annotation can be found in Data S3 and updated Data S15. XLOC_193638 was identified as a **MEG from endosperm** at E8 stage between AxS and SxA.*

*XLOC_116226;chr4B:660904503-660911006, corresponds to TraesCS4B02G381000 in reference genome. The annotation can be found in Data S3 and updated Data S15. XLOC_116226 was identified as a **PEG from embryos** at E3 stage between AxS and SxA.*

*XLOC_210220;chr7B:138877164-138887015, corresponds to TraesCS7B02G119800 in reference genome. The annotation can be found in Data S3 and updated Data S15. XLOC_210220 was identified as a **PEG from endosperm** at E8 stage between AxS and SxA.*

As suggested, we have added the chromosomal location of the selected genes in Figure 7, updated Data S15.

Relevant description in the main text as detailed below:

line 483-486: “The expression levels of three DEGs, XLOC_061322 on chr2D, XLOC_123600 on chr4D and XLOC_114665 on chr4B in embryos at stage E4, E5 and E6 from the cross between Strong Field and AC Barrie (SxA) were quantified and compared (Figure 7A). ”

Line 498-501, “Selected imprinted genes between SxA and AxS were confirmed in embryo and endosperm. Two genes, XLOC_045394 on chr2B and XLOC_116226 on chr4B, were imprinting genes from embryos at E3 stage. XLOC_193638 on chr7A and XLOC_210220 on chr7B were identified as imprinting genes from endosperm at E8 stage. ”

Reviewer #3 (Remarks to the Author):

The authors have answered all the questions and addressed the concerns I had during the initial review, and they have modified the manuscript accordingly. I have no further comments. I congratulate them on their achievement to improve the report. I am confident the manuscript has merit for publication in the ‘Communications Biology’ and should now be accepted for publication.

***Response:** We thank this reviewer for critical assessment and approval of our revised manuscript. The reviewer’s constructive feedback has improved the quality of our manuscript.*

REVIEWERS' COMMENTS:

Reviewer #2 (Remarks to the Author):

The revised manuscript has addressed all my comments. Thus, I have no further comment for the resubmitted manuscript.